# ADAPTIVE ROBUST EVIDENTIAL OPTIMIZATION FOR OPEN SET DETECTION FROM IMBALANCED DATA

**Hitesh Sapkota & Qi Yu**
Rochester Institute of Technology
`{hxs1943, qi.yu}@rit.edu`

## ABSTRACT

Open set detection (OSD) aims at identifying data samples of an unknown class (*i.e.,* open set) from those of known classes (*i.e.,* closed set) based on a model trained from closed set samples. However, a closed set may involve a highly imbalanced class distribution. Accurately differentiating open set samples and those from a minority class in the closed set poses a fundamental challenge as the model may be equally uncertain when recognizing samples from the minority class. In this paper, we propose Adaptive Robust Evidential Optimization (AREO) that offers a principled way to quantify sample uncertainty through evidential learning while optimally balancing the model training over all classes in the closed set through adaptive distributively robust optimization (DRO). To avoid the model to primarily focus on the most difficult samples by following the standard DRO, adaptive DRO training is performed, which is governed by a novel multi-scheduler learning mechanism to ensure an optimal model training behavior that gives sufficient attention to the difficult samples and the minority class while capable of learning common patterns from the majority classes. Our experimental results on multiple real-world datasets demonstrate that the proposed model outputs uncertainty scores that can clearly separate samples from closed and open sets, respectively, and the detection results outperform the competitive baselines.

## 1 INTRODUCTION

In many practical scenarios (*e.g.,* drug discovery, anomaly detection *etc.*), it is likely to encounter unknown samples and it is desirable that the model can properly detect these samples as *unknown*. Various approaches have been proposed to tackle the unknown sample detection problem (Bendale & Boult, 2016; Sun et al., 2020), using techniques such as Weibull-Calibration SVM (W-SVM) (Scheirer et al., 2013), reconstruction error (Zhang & Patel, 2017), nearest neighbor (Júnior et al., 2016), and quasi-linear function (Cevikalp & Yavuz, 2017). As a representative example, the Openmax framework removes softmax from the last layer of a neural network and includes an additional layer to produce the probability of a sample being unknown. This essentially redistributes the probability mass to $(K + 1)$ classes (with unknown being a new class). Multiple efforts follow this direction (Sun et al., 2020; Neal et al., 2018). While this technique is viable to detect open-set samples, the additional layer is included during the testing phase. As a result, the training still follows the closed set assumption.

Recent advances in uncertainty quantification provide a more systematic way to break the closed set limitation by explicitly modeling the uncertainty mass that corresponds to the unknown class. One representative work is the evidential deep learning (EDL) model (Sensoy et al., 2018), which treats the predicted multi-class probability as a multinomial opinion according to subjective logic (Jøsang, 2016). Similar to EDL, Prior Networks (PNs) (Malinin & Gales, 2018) explicitly considers the distributional uncertainty that quantifies the distributional mismatch (Malinin & Gales, 2018). The Posterior Networks further improves PNs by leveraging normalizing flows for density estimation in the latent space to predict a posterior distribution, which can be used to identify out-of-distribution (OOD) samples from in-distribution ones (Charpentier et al., 2020).

Despite the promising progress in OSD that focuses on differentiating samples from the closed and open sets, respectively, limited attention has been devoted to the situation where the closed set involves highly imbalanced classes, which may be quite common in many practical settings. For

example, for anomaly detection, the known types of anomalies available for model training are usually unevenly distributed into multiple categories (*e.g.,* car accident vs. shooting). Similarly, for computer-aided medical diagnosis, the known diseases (to the model) may be highly imbalanced based on the available cases. Thus, following the standard Empirical Risk Minimization (ERM) framework for training, the model may not learn properly from the minority class due to the lack of positive samples. As a result, it is more likely to misidentify a minority-class sample as an unknown-class sample during OSD, leading to a high false-positive rate.

Distributionally Robust Optimization (DRO) offers an effective means to handle the imbalance class distribution in the closed set setting (Qi et al., 2020; Zhu et al., 2019). In DRO, the worst case weighted loss is optimized, where the weights are searched in a given neighborhood (referred to as the uncertainty set) of the empirical sample distribution such that the overall loss is maximized. By expanding the uncertainty set, the model is encouraged to assign higher weights to difficult samples. As a result, samples from the minority class will be given more emphasis during model training if not properly learned (which incurs a larger loss). Another common solution to handle imbalanced class distribution in the closed set is through oversampling to achieve a more balanced class distribution (Chawla et al., 2002). While both oversampling and DRO may help to improve the closed set performance, neither of them is adequate to address OSD from imbalanced data.

A fundamental challenge lies in the interplay between samples from the minority class and the difficult samples from the majority classes. As a result, simply oversampling the minority class may neglect these difficult samples. Similarly, applying DRO with a flexible uncertainty set may put too much emphasis on these difficult samples and ignore the minority class as well as some representative samples from the majority classes, which affects proper model training. In fact, directly applying these models for OSD may lead to even worse detection performance, which is evidenced by our experimental results. Few recent approaches try to address the OSD under class-imbalanced setting. Liu et al. (2019) leverage the visual similarity across the centroids of closed set classes to allow more effective training from the minority class samples. However, it is possible that the samples from the minority class may look quite different from most other samples, making such a strategy less effective. Further, Wang et al. (2022) try to push minority class samples away from open set ones in the feature space using contrastive learning. However, the final OSD depends heavily on the selection of open set samples as evidenced by our experiment results.

To systematically tackle the fundamental challenge as outlined above, we propose Adaptive Robust Evidential Optimization (AREO) that offers a principled way to quantify sample uncertainty through evidential learning while optimally balancing the model training over all classes in the closed set through novel adaptive DRO learning. To avoid the model from primarily focusing on the most difficult samples by following the standard DRO, the adaptive learning strategy gradually increases the size of the uncertainty set using Multi Scheduler Function (`MSF`), which allows the model to learn from easy to hard samples. A class-ratio biased loss is further assigned to the minority class to ensure proper learning from its limited samples. Our main contribution is fourfold:

- a novel extension of DRO to evidential learning, which enables principled uncertainty quantification under the class imbalanced setting, critical for many applications, including OSD,
- adaptive DRO training governed by a uniquely designed multi-scheduler learning mechanism to ensure an optimal model training behavior that gives sufficient attention to the difficult samples and the minority class while capable of learning common patterns from the majority classes,
- theoretical connection to a boosting model (*i.e.,* AdaBoost), which ensures the nice convergence and generalization properties of AREO,
- state-of-the-art OSD performance on various datasets.

## 2 RELATED WORK

**Open set detection.** Various SVM based techniques (Scheirer et al., 2013; Jain et al., 2014; Scheirer et al., 2014) have been proposed for OSD. For instance, Scheirer et al. (Scheirer et al., 2013) proposed an SVM based model, which performs detection using a Weibull-calibrated SVM (W-SVM) by leveraging Extreme Value Theory (EVT). Reconstruction based approaches have been proposed (Zhang & Patel, 2017), where a threshold defined over the reconstruction error is used to decide whether the sample is from a known or an unknown class. Other traditional models, such as nearest neighbor (Júnior et al., 2016), quasi-linear function (Cevikalp & Yavuz, 2017), have also been explored as well. Deep learning models have been increasingly applied for open

set detection (Yoshihashi et al., 2019; Sun et al., 2020; Bendale & Boult, 2016). As an example, OpenMAX replaces the softmax function and probability of the softmax is redistributed to produce the probability of a sample being unknown (Bendale & Boult, 2016). Sun et al. (2020) proposed VAE based open set recognition, where the probability of a sample belonging to each of the known classes is used as a proxy to detect whether the sample is known or unknown. Each known class distribution is modeled as a Gaussian using the training data. Some recent approaches aim to learn a more compact representation of closed set samples (Cevikalp et al., 2021; Yang et al., 2020) or push the open set class samples to a specific region in an embedding space for better recognition (Chen et al., 2021). Special loss functions (Dhamija et al., 2018) and generative processes (Perera et al., 2020) have also been leveraged to separate open set samples from closed set ones.

Recently, systematic approaches have been presented to break the closed set limitation by explicitly modeling the uncertainty mass belonging to the unknown distribution. One of the representative work inline with this is the evidential deep learning (EDL) model (Sensoy et al., 2018). Similar to this work, Malinin & Gales (2018) propose Prior Networks (PNs) that explicitly consider the distributional uncertainty to quantify the distribution mismatch. Despite having a natural way to quantify the uncertainty, both of these methods require OOD data samples for model training, which is less practical. Charpentier et al. (2020) propose the posterior networks that leverage the normalizing flows for density estimation in the latent space in order to predict the posterior distribution by only using in-distribution samples. Despite the significant progress in OSD, limited attention has been drawn to the scenario, where the closed set involves highly imbalanced classes. Few recent works try to tackle this fundamental challenge of OSD under class-imbalanced setting. Liu et al. (2019) propose a technique based on the assumption that visual similarity exists between head and tail classes in the closed set. A model is designed to leverage this similarity to make it more robust for recognizing minority class samples. However, such an assumption may not universally hold, which limits the applicability of the model in general settings. Further, Wang et al. (2022) leverage contrastive learning to push the minority class samples away from the open set ones in the feature space during the training process. However, the final OSD performance highly depends on the training open set samples.

**Distributionally robust optimization.** Distributionally robust optimization is based on principled statistical learning theory, where the worst case weighted loss is optimized by searching the weights in a given uncertainty set (Duchi & Namkoong, 2019; Zhu et al., 2019; Namkoong & Duchi, 2016). DRO offers a systematic way to handle the imbalanced class distribution and has been commonly used in supervised learning setting (Qi et al., 2020; Zhu et al., 2019) as well as in multiple instance learning (Sapkota et al., 2021). In a similar way, Li et al. (2020) propose a technique called Tilted Empirical Risk Minimization (TERM) by redefining the ERM with the introduction of hyperparameter $t$. Depending on the tunable parameter $t$ value, different variants of loss (maximum, minimum, and average) are recovered and thereby provide a unified way to perform effective training in the presence of outlier and class imbalance scenarios. While DRO may help to improve the closed set performance, it is not sufficient to address the OSD problem with imbalanced data. This is because DRO with a flexible uncertainty set may put too much emphasis on the difficult samples and ignores the ones from the minority class as well as representative samples from majority classes.

Our proposed AREO model offers an adaptive learning strategy to learn from easy samples in the early training phase and gradually shift the focus to the difficult samples. Furthermore, the class-ratio biased loss ensures proper learning from the limited samples in the minority class.

## 3 METHODOLOGY

### 3.1 PRELIMINARIES

**Evidential Learning for OSD.** Let $\mathcal{D}_N = \{\mathbf{X}, \mathbf{Y}\} = \{(\mathbf{x}_1, \mathbf{y}_1), ..., (\mathbf{x}_N, \mathbf{y}_N)\}$ be a set of training samples in the closed set. Each $\mathbf{x}_n \in \mathbb{R}^D$ is a $D$-dimensional feature vector and $\mathbf{y}_n \in \{0,1\}^C$ indicates the one hot encoding associated with its class label: $y_{nj} = 1$ and $y_{nk} = 0$ for all $k \neq j$ with $j$ being the true label. Following the principle of Subjective Logic (SL) (Jøsang, 2016), we consider a total of $C + 1$ mass values with $C$ being the number of classes. We assign a belief mass $b_c, \forall c \in [C]$, to each singleton, which corresponds to one class in the closed set and the remaining mass is referred to as the uncertainty mass, denoted by $u$. The belief masses and the uncertainty mass are all non-negative and sum to one: $u + \sum_{c=1}^{C} b_c = 1, u \geq 0$ and $b_c \geq 0$. They can be evaluated as

$$b_c = \frac{e_c}{S}, \quad u = \frac{C}{S} \qquad (1)$$

where $S = \sum_{c=1}^{C}(e_c + 1)$ with $e_c \geq 0$ being the evidence derived for the $c^{th}$ singleton, which can be generated by a neural network enabled with a non-negative output. The belief mass assignment in the above expression corresponds to a Dirichlet distribution with the concentration parameters $\alpha_c = e_c + 1$:

$$\text{Dir}(\mathbf{p}|\boldsymbol{\alpha}) = \begin{cases} \frac{1}{B(\boldsymbol{\alpha})} \prod_{c=1}^{C} p_c^{\alpha_c - 1}, & \text{for } \mathbf{p} \in \mathcal{S}_C \\ 0, & \text{otherwise} \end{cases} \quad (2)$$

where $\mathcal{S}_C$ is a $(C-1)$-simplex and $B(\alpha)$ is a beta function. Given the evidences, the expected probability for the $c^{th}$ singleton is given by $\mathbb{E}[p_c] = \frac{\alpha_c}{S}$. Consider a sample $\mathbf{x}_n$ and let $\mathbf{f}(\mathbf{x}_n, \Theta)$ denote the evidence vector generated by an evidential neural network parameterized by $\Theta$. This allows us to fully characterize the Dirichlet distribution, whose mean vector gives rise to the probability of assigning $\mathbf{x}_n$ to each class. There are multiple ways to design a loss function to train the evidential neural network (Sensoy et al., 2018). A simple but effective option is the sum of square loss:

$$rl_n^{\text{EL}}(\Theta) = \|\mathbf{y}_n - \mathbb{E}[\mathbf{p}_n]\|_2^2 + \lambda_t KL[\text{Dir}(\mathbf{p}_n|\tilde{\boldsymbol{\alpha}}_n)|\text{Dir}(\mathbf{p}_n|(1, ...., 1)^{\top})] \quad (3)$$

where $\lambda_t = \min(1, \frac{t}{10})$ is the annealing coefficient at epoch $t$ and $\tilde{\boldsymbol{\alpha}}_n = \mathbf{y}_n + (1 - \mathbf{y}_n) \odot \boldsymbol{\alpha}_n$.

**Remark.** Besides being used as a powerful model for closed set classification, a unique benefit of evidential learning is that it offers a principled way to quantify the uncertainty mass, which is explicitly allocated to account for something that is 'unknown' to the model. Intuitively, a properly trained evidential model will output a high total evidence for data samples whose features are sufficiently exposed to the model during training. In contrast, it should predict a low total evidence for less representative samples in the training data. For these samples, their corresponding uncertainty mass $u$ will be large (as the total mass sums to one). As a result, the uncertainty mass fits squarely for detecting open set samples, which have not been exposed to the model that is trained using the closed set samples.

**Distributionally Robust Optimization.** Distributionally Robust Optimization (DRO) is inherently used to handle the minority and/or difficult class samples by optimizing the worst-case loss where weights assigned to each sample are given by uncertainty set. Let $l_n(\Theta)$ be the loss for the $\mathbf{x}_n$ sample network parameterized by $\Theta$. Then the corresponding DRO loss is given as

$$\mathcal{L}^{\text{DRO}}(\Theta) = \max_{\mathbf{p} \in \mathcal{P}^{\text{DRO}}} \sum_{n=1}^{N} p_n l_n(\Theta) \quad (4)$$

The uncertainty set defined to assign weights ($\mathbf{p}$) is given as

$$\mathcal{P}^{\text{DRO}} := \left\{ \mathbf{p} \in \mathbb{R}^N : \mathbf{p}^{\top}\mathbb{1} = 1, \mathbf{p} \geq 0, D_f(\mathbf{p}\|\frac{\mathbb{1}}{N}) \leq \eta \right\} \quad (5)$$

where $D_f(\mathbf{p}\|\mathbf{q})$ is $f$-divergence between two distributions $\mathbf{p}$ and $\mathbf{q}$ and $\eta$ controls the size of the uncertainty set. When $\eta$ is large, the weight distribution $\mathbf{p}$ can deviate a lot from the uniform distribution, making it possible to assign a very high weight to certain data samples. In contrast, a small $\eta$ will constrain $\mathbf{p}$ to be close to the uniform distribution and all samples share a similar weight.

### 3.2 DISTRIBUTIONALLY ROBUST EVIDENTIAL OPTIMIZATION

The standard evidential learning does not explicitly consider an imbalanced class distribution. Further, it also does not focus on the difficult samples resulting from multi-modality where a single class can have multiple types of samples. As a result, minority classes and/or difficult samples are usually assigned a higher uncertainty mass due to a lack of sufficient training data. While this may not significantly impact the closed set performance (*i.e.,* accuracy), it poses a more severe issue for OSD as difficult/minority class samples become equally uncertain as those open set samples. To address this challenging solution, one straightforward way would be to integrate evidential learning with DRO for robust uncertainty mass quantification on minority class/difficult samples in the close-set. Intuitively, since the model explicitly focuses on learning from minority class/difficult samples, it provides a low uncertainty mass for minority/difficult samples while remain high (in terms of uncertainty mass) for those open set samples. This novel integration of DRO and evidential learning allows us to define a distributionally robust evidential loss (DREL) given as

$$\mathcal{L}^{\text{DREL}}(\Theta) = \max_{\mathbf{p} \in \mathcal{P}^{\text{DRO}}} \sum_{n=1}^{N} p_n l_n^{\text{EL}}(\Theta) \quad (6)$$

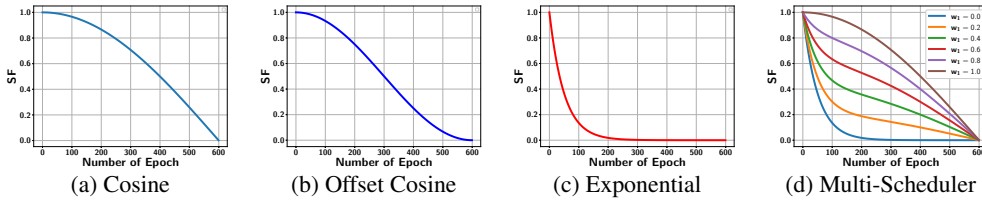

Figure 1: Examples of Scheduler Functions

Details of solving (6) is provided in Appendix B. Depending on $\eta$ in the uncertainty set, we can decide whether we want to assign an equal weight to all data samples or focus on the most difficult ones. The lemma below reveals the relationship between DREL and the standard evidential loss.

**Lemma 1.** *With $\eta \to 0$, the EDL loss under DRO reduces to the standard EDL loss.*

When $\eta$ is set to be very small, the model gives similar weights to all samples, which allows them to participate equally in the training process. At another extreme, we can direct the model to fully focus on the most difficult sample with the maximum loss, as summarized in the lemma below.

**Lemma 2.** *With $\eta \to \infty$, the loss under DRO becomes equivalent to a maximum loss based approach focusing only on the hardest sample.*

The above lemma implies that a highly flexible uncertainty set may cause the model to put too much emphasis on difficult samples. Since these difficult samples may come from the majority classes, simply setting a large $\eta$ will not be necessary to direct the model's attention to the samples from the minority class. Furthermore, using a flexible uncertainty set in the initial phase of the model training may misguide the model to neglect a large number of representative data samples. As a result, the model will not be able to capture the common patterns that exhibit in most of the training samples. As such, the direct integration of DRO and EDL does work well which is also justified experimentally through the comparison of the proposed technique with DRO technique.

### 3.3 ADAPTIVE ROBUST EVIDENTIAL OPTIMIZATION (AREO)

The key idea to address the limitations in the distributionally robust evidential optimization is to gradually increase the size of the uncertainty set, which allows the model to learn from easy to hard samples from closed set classes. Scheduler functions (SF) provide a natural way to achieve the desired training behavior. Figure 1 (a-c) shows three typical SFs, including cosine in (a): $\cos\left(\frac{\pi t}{2T}\right)$, offset cosine in (b): $\frac{1}{2}\cos\left(\frac{\pi t}{T}\right) + \frac{1}{2}$, and exponential in (c): $\exp\left(-\frac{t}{\beta}\right)$, where $t$ denotes the index of the training epoch, $T$ is the terminating epoch, and $\beta$ is a specific parameter of the exponential function. It can be seen that while the general trends of different SFs are similar, they exhibit some key differences that may lead to quite distinct model training behaviors. For example, a cosine function can help to ensure the uncertainty set to stay small for a relatively longer time in the beginning of model training. This ensures the model to learn from the representative samples in the majority classes (according to Lemma 1). In contrast, an exponential function can change the size of the uncertainty set very rapidly, which can give the model more time to learn from the difficult samples at the later phase (according to Lemma 2). The offset cosine function can offer both a relatively long initial learning and later learning phases. However, choosing a SF that best matches the nature of a given dataset poses a key challenge. Furthermore, a single SF may not be rich enough to express the desired training behavior of a complex dataset.

To address this key challenge, we propose to conduct multi-scheduler learning to automatically construct a composite scheduler function that can be automatically learned for each given dataset to deliver the optimal training behavior. More specifically, the multi-scheduler function (MSF) is formulated as a convex combination of a set of atomic SFs:

$$\text{MSF}(\mathbf{w}, \boldsymbol{\beta}, t, T) = \sum_{m=1}^{M} w_m \text{SF}_m(\beta_m, t, T), \sum_{m=1}^{M} w_m = 1, w_m \geq 0 \quad \forall m \in [M] \qquad (7)$$

where $\mathbf{w}$ are the mixing weights and $\boldsymbol{\beta}$ is a set of specific parameters for the atomic SFs. Figure 1 (d) visualizes an example MSF that combines a cosine and exponential functions with different mixing weights and fixed $\beta = 20, T = 600$. As can be seen, the MSF is much more expressive then either its component SF, which makes it capable to represent a much broader range of training behaviors.

By leveraging the proposed MSF to control the size of the uncertainty set, we can achieve adaptive robust training. Let $\eta_0$ be the initial size of the uncertainty set and the size of the set at epoch $t$ is

$$\eta_t = \frac{\eta_{t-1}}{\text{MSF}(\mathbf{w}, \boldsymbol{\beta}, t, T)} \tag{8}$$

Based on this adaptive uncertainty set, we define the Adaptive Robust Evidential Loss (AREL) as

$$\mathcal{L}^{\text{AREL}}(\Theta) = \max_{p \in \mathcal{P}^{\text{ARO}}} \sum_{n=1}^{N} p_n l_n^{\text{EL}}(\Theta) \tag{9}$$

where $l_n^{\text{EL}}$ is the uncertainty mass loss for the $\mathbf{x}_n$ given by Eq. (3) under adaptive robust optimization framework and $\mathcal{P}^{\text{ARO}}$ is the adaptive robust uncertainty set defined as

$$\mathcal{P}^{\text{ARO}} := \left\{ \mathbf{p} \in \mathbb{R}^N : \mathbf{p}^\top \mathbb{1} = 1, \mathbf{p} \geq 0, D_f(\mathbf{p} \| \frac{\mathbb{1}}{N}) \leq \eta_t \right\} \tag{10}$$

As $\eta_t$ increases, the model gradually shifts its focus from easier samples to the more difficult ones. In this way, the model can be trained to first capture the common patterns in the data and then conduct fine-tuning by attending to those difficult samples. However, for imbalanced classes, there may be a good number of difficult samples from the majority classes. Therefore, solely controlling the size of the uncertainty set does not guarantee a sufficient training over the minority class. To address this, we further leverage the label of the minority-class $c$ to formulate a ratio biased weight augmentation on samples from this class. Let $p(c) = \sum_{\forall y_{nc}=1} p_n$ be the total weight for minority class $c$ obtained by solving (9). Then, the weights for the minority class samples are adjusted as:

$$\widetilde{p(c)} = \begin{cases} p(c), \text{if } p(c) \geq \frac{1}{C} \\ \min\left(\frac{1}{C}, p(c)^{\text{MSF}(\mathbf{w}', \boldsymbol{\beta}', t, T)}\right), \text{otherwise} \end{cases} \qquad \widetilde{p_n} = \begin{cases} \frac{\widetilde{p(c)}}{p(c)} p_n, \text{if } y_{nc} = 1 \\ \frac{1 - \widetilde{p(c)}}{1 - p(c)} p_n, \text{otherwise} \end{cases} \tag{11}$$

As the MSF monotonically decreases over the training epochs, the total weight for the minority class samples will eventually reach $\frac{1}{C}$, making it equally weighted as the other $(C-1)$ classes.

**Remark.** Our approach considers a minority class if there is an obvious gap between the percentage of samples from the minority class over the total samples from all C classes and $\frac{1}{C}$. Any other class that is not a minority one is regarded as a majority class. Our approach can handle the multiple minority classes which can be achieved by applying the ratio biased weighted augmentation (given by Eq. (11)) to each minority class.

The adaptive robust training is achieved through a bi-level optimization, where the inner loop optimizes the the model parameters ($\Theta$) and the outer loop optimizes the MSF parameters $\mathbf{W} = \{\mathbf{w}, \mathbf{w}', \boldsymbol{\beta}, \boldsymbol{\beta}'\}$:

$$\min_{\mathbf{W}} \mathcal{L}_{val}^{\text{AREL}}(\Theta^*, \mathbf{W}), \text{ s.t. } \Theta^* = \arg\min_{\Theta} \mathcal{L}_{train}^{\text{AREL}}(\Theta, \mathbf{W}) \tag{12}$$

where $\mathcal{L}_{train}^{\text{AREL}}, \mathcal{L}_{val}^{\text{AREL}}$ are training and validation losses, respectively. The outer loop optimization can be solved by computing the Hypergradients (Maclaurin et al., 2015; Pedregosa, 2016) or through a population-based methods (Jaderberg et al., 2017), where the former may easily get stuck in local optimum (Tao et al., 2020). To this end, we extend the existing population based method to learn an optimal MSF and the details are given in Appendix B.

## 3.4 THEORETICAL ANALYSIS

We establish the key theoretical properties of AREO, including the convergence speed in model training and the generalization capability by formally demonstrating the equivalence between AREO and AdaBoost under a non-convex robust uncertainty loss. The key idea is to leverage the equivalence between AdaBoost and the gradient descent search of an optimal function from a linear combination of a set of (weak) learners (Mohri et al., 2012; Blanchet et al., 2019). Let $\mathcal{F} = \{f_1, ..., f_K\}$ be a set of different classifiers, and the linear span generated by the set $\mathcal{F}$ is

$$\text{LS}(\mathcal{F}) = \left\{ f : f = \sum_{k=1}^{K} \sigma_k f_k, 1 \leq k \leq K \right\} \tag{13}$$

AREO training consists of two alternative updates between optimizing the worst case probability and predicting function $f$. The update in function prediction can be regarded as finding a sub-gradient $\mathcal{G}_t \in \partial \mathcal{L}^{\text{AREL}}(f_t)$ and updating with $\prod_{LS(\mathcal{F})}(\mathcal{G}) = \arg\min_{f \in LS(\mathcal{F})} \|f - \mathcal{G}_t\|_{\mathcal{D}_N}$ where $\mathcal{D}_N$ is the training data. Letting $\mathcal{L}_n(f_t)$ be the loss associated with the data sample $\mathbf{x}_n$, the update of $\mathbf{p}$ involves the optimization of the following objective with $f_t$ being fixed:

$$\mathcal{L}^{\text{AREL}}(f_t) = \max_{\mathbf{p} \in \mathcal{P}^{\text{ARO}}} \sum_{n=1}^{N} p_n \mathcal{L}_n(f_t) \tag{14}$$

where the uncertainty set is given by (10). The corresponding Lagrangian of the above optimization problem is given by

$$\max_{\mathbf{p} \geq 0, \mathbf{p}^\top \mathbb{1} = 1} \sum_{n=1}^{N} p_n \mathcal{L}_n(f_t) - \alpha \left[ \left( \sum_{n=1}^{N} p_n \log p_n \right) - \eta_t \right] \tag{15}$$

It should be noted that finding the optimal $f$ value is non-trivial because the optimization involves the nonconvex loss (*i.e.,* $\mathcal{L}^{\text{AREL}}$). This creates difficulty showing equivalence between AREO and AdaBoost. To ensure the convergence of $f$ to a stationary point, we adapt the ProbAbilistic Gradient Estimator technique (PAGE) (Li et al., 2021) to our unique adaptive robust evidential optimization setting. This convergence guarantee helps to move forward showing the equivalence between AREO and Adaboost given by the theorem below.

**Theorem 3.** *Under the assumption of finite exponential moment for $\mathcal{L}_n(f)$, with $\alpha \geq 0$ being sufficiently large and*

$$\eta_t = \beta^* \psi^{'}(\beta^*) - \psi(\beta^*) \tag{16}$$

*the worst case probability $\mathbf{p}^*$ is given by*

$$p_n^* = \frac{\exp\left(\frac{\mathcal{L}_n(f_t)}{\alpha}\right)}{\sum_{j=1}^{N} \exp\left(\frac{\mathcal{L}_j(f_t)}{\alpha}\right)} \tag{17}$$

*where $\beta^* = \frac{1}{\alpha^*}$, $\alpha^* \geq 0$ be the optimal $\alpha$, and $\psi(\beta) = \log\left[\frac{\sum_{n=1}^{N} \exp(\beta \mathcal{L}_n(f_t))}{N}\right]$. The alternative optimization between $f$ and with above worst case probability solution exactly recovers the AdaBoost algorithm proposed in (Freund & Schapire, 1997).*

**Remark.** There are several key benefits of connecting AREO with AdaBoost. First, AdaBoost is less prone to overfitting even running for a large number of iterations (Mease & Wyner, 2008). Inheriting such a property is crucial for OSD as an overfitted evidential model can produce highly confident wrong predictions. This implies that a low uncertainty may be predicted for samples that the model is less familiar with, resulting in a false negative detection of an open set sample. Furthermore, since the target function is expressed as a linear combination of a set of weak learners, the optimal function can be regarded as maximizing the $l_1$ geometric margin among the training samples to ensure good generalization capability like other maximum-margin classifiers (Mohri et al., 2012). This ensures a decent closed set performance from AREO (as shown by our experiments). The proof of Theorem 3 is provided in Appendix C.

## 4 EXPERIMENTS

We perform extensive experimentation to evaluate the effectiveness of the proposed AREO model. We first describe five real-world image datasets where a minority class is introduced to create an imbalanced setting. We then assess the OSD performance of the proposed technique by comparing with competitive baselines. Finally, we conduct some qualitative analysis, which uncovers deeper insights on the performance advantage of the proposed model.

### 4.1 DATASETS

Our experiments involve five real-world image datasets: Cifar10, Cifar100 (Krizhevsky, 2009), ImageNet (Deng et al., 2009), MNIST (Deng, 2012), and Architecture Heritage Elements Dataset (AHED) (Llamas, 2017). In our experimentation, model training is performed solely based on the closed set samples. During the detection phase, the testing samples corresponding to the closed

set classes will be assessed against the samples from open set classes. For all datasets, for the hyperparameter optimization, randomly selected 20% of the training set is used. The brief description for each dataset is given below. For the detailed description and data sample distribution in majority and minority classes, please refer to the Appendix.

- **MNIST**: Five classes are treated as open set and the rest as the closed set. To make the dataset imbalanced, we consider class '3' as a minority class and randomly select 30% data samples as compared with other majority classes. The same imbalanced ratio is applied to both training and testing sets. In addition to the MNIST open set classes as described above, we follow other existing works (Sun et al., 2020) and further test the OSD performance on additional open set samples from three more sources: (1) MNIST-Noise, (2) Noise, and (3) Omnigolot (Lake et al., 2015).
- **Cifar10**: Five classes are assigned as open set and closed set, respectively. Bird is made as the minority class using the same strategy introduced above. In addition to the open set classes from Cifar10 itself, we further assess the OSD performance with Cifar+10 and Cifar+50.
- **Cifar100**: 'Living being' related super classes are assigned as the closed set and the remaining super classes are assigned as the open set. We make 'insect' related classes as the minority one.
- **ImageNet**: Five classes are assigned as open set and closed set, respectively. We make 'king crab' as the minority class.
- **Architectural Heritage Elements Dataset (AHED)**: Five classes are assigned as open set and closed set, respectively. This is inherently highly imbalanced dataset where number of data points are unevenly distributed across different classes. The class 'portal' is the minority one.

## 4.2 EXPERIMENTAL SETTINGS

**Evaluation metric.** To assess the model performance, we report mean average precision (MAP) score which summarizes the precision-recall curve as a weighted mean of precision achieved at each threshold, with the increase in recall from previous threshold as the weight. Specifically, in the OSD, we treat the open set samples as positive and closed set samples as negative and compute the MAP score based on the uncertainty score produced by the trained model. Different from AUROC, MAP places more emphasis on initial part of the ROC curve, which gives preference if model can rank the open set samples on the top based on their predicted uncertainty scores. This MAP metric works well in practice as the main focus may be devoted to the first few predicted candidate samples, especially when there is a long candidate list. The theoretical result shows that MAP is approximately the AUROC times the initial precision of the model (Su et al., 2015). Therefore, we focus on reporting the MAP performance and leave the AUROC results in Appendix D. It is worth to note that our AUROC results also show a consistent trend as the MAP results.

**Network architecture.** In terms of the architecture of the evidential neural network, for all datasets, we use an LeNet5 network with tanh activation in the feature extractor and ReLU in the fully connected layers. For training, we use the Adam optimizer with a learning rate of 0.001 and $l_2$ regularization with a coefficient of 0.001. The detailed hyperparameter setting is provided in Appendix.

## 4.3 PERFORMANCE COMPARISON

In our comparison study, we include baselines that are most relevant to our model, including EDL, EDL augmented with oversampling using SMOTE (Chawla et al., 2002) (referred to as AEDL), and EDL with standard DRO training (referred to as DRO). Further, we also compare the performance with the Posterior networks (Charpentier et al., 2020) and its robust form, PostNet (RS), proposed by Kopetzki et al. (Kopetzki et al., 2021). In addition, we also compare with representative baselines with outstanding OSD performance: OpenMAX (Bendale & Boult, 2016), CGDL (Sun et al., 2020), and OLTR (Liu et al., 2019). Please refer to the Appendix D for the more detailed description of the baselines used in our comparison study along with additional results and an ablation study.

Tables 1 presents the OSD performance comparison between different models for all five datasets. AREO consistently outperforms all the baselines across all the datasets. For certain datasets, the performance advantage over the second best model is more than or close to 10%. This clearly demonstrates the benefits of conducting evidential learning through adaptive DRO training to achieve an optimal balanced learning from all classes and different types of data samples. We also observe that EDL consistently performs better than other non-evidential learning based models, such as OpenMAX, in most cases. The better OSD performance from EDL is attributed to its explicit modeling of the uncertainty mass that works naturally for detecting the open set samples. In contrast, directly applying DRO with a flexible uncertainty set, which aims to address the imbalanced class

Table 1: OSD (MAP) performance on all datasets

| Approach | Cifar10 | | | Cifar100 | ImageNet |
|---|---|---|---|---|---|
| | Cifar10 | Cifar+10 | Cifar+50 | | |
| EDL | $62.42 \pm 0.31$ | $29.23 \pm 0.38$ | $65.75 \pm 1.11$ | $52.00 \pm 2.40$ | $55.93 \pm 4.30$ |
| AEDL | $54.19 \pm 0.77$ | $26.21 \pm 0.68$ | $63.04 \pm 0.70$ | $52.79 \pm 0.91$ | $57.94 \pm 0.07$ |
| DRO | $57.86 \pm 2.94$ | $18.35 \pm 0.40$ | $56.04 \pm 1.63$ | $50.78 \pm 4.44$ | $55.67 \pm 3.86$ |
| OpenMAX | $59.65 \pm 1.03$ | $24.48 \pm 1.34$ | $62.80 \pm 2.08$ | $50.88 \pm 0.60$ | $53.24 \pm 0.39$ |
| CGDL | $54.27 \pm 2.06$ | $16.83 \pm 0.20$ | $50.15 \pm 1.08$ | $50.59 \pm 4.56$ | $55.47 \pm 1.53$ |
| PostNet | $56.71 \pm 6.08$ | $25.71 \pm 5.39$ | $62.51 \pm 4.68$ | $53.85 \pm 2.76$ | $56.83 \pm 1.52$ |
| PostNet (RS) | $51.54 \pm 11.32$ | $18.28 \pm 1.50$ | $53.13 \pm 4.33$ | $51.75 \pm 0.98$ | $56.21 \pm 0.75$ |
| OLTR | $56.37 \pm 0.25$ | $19.59 \pm 0.49$ | $53.98 \pm 0.68$ | $48.48 \pm 0.29$ | $50.71 \pm 0.66$ |
| **AREO** | $\mathbf{72.48 \pm 4.08}$ | $\mathbf{37.14 \pm 2.06}$ | $\mathbf{73.87 \pm 1.42}$ | $\mathbf{57.52 \pm 1.60}$ | $\mathbf{62.02 \pm 1.11}$ |
| Approach | MNIST | | | | AHED |
| | MNIST | Noise | MNIST-Noise | Omnigolot | |
| EDL | $87.32 \pm 4.01$ | $82.16 \pm 8.74$ | $82.89 \pm 8.06$ | $77.62 \pm 6.79$ | $50.23 \pm 1.84$ |
| AEDL | $75.37 \pm 11.14$ | $71.90 \pm 11.45$ | $76.23 \pm 12.67$ | $67.29 \pm 10.77$ | $52.22 \pm 0.26$ |
| DRO | $63.25 \pm 4.32$ | $46.78 \pm 1.22$ | $49.59 \pm 3.98$ | $48.15 \pm 1.70$ | $42.28 \pm 0.18$ |
| OpenMAX | $84.11 \pm 1.55$ | $83.03 \pm 1.71$ | $78.31 \pm 2.745$ | $81.14 \pm 0.89$ | $48.13 \pm 0.19$ |
| CGDL | $61.33 \pm 1.53$ | $74.88 \pm 8.42$ | $73.92 \pm 8.41$ | $90.72 \pm 2.16$ | $48.57 \pm 1.39$ |
| PostNet | $55.58 \pm 9.12$ | $47.53 \pm 13.88$ | $43.94 \pm 8.00$ | $72.79 \pm 4.24$ | $46.69 \pm 1.90$ |
| PostNet (RS) | $49.3 \pm 4.071$ | $36.14 \pm 0.59$ | $39.96 \pm 2.20$ | $77.43 \pm 9.80$ | $46.10 \pm 4.37$ |
| OLTR | $86.38 \pm 0.51$ | $90.61 \pm 1.43$ | $83.75 \pm 1.28$ | $55.27 \pm 3.04$ | $49.26 \pm 1.66$ |
| **AREO** | $\mathbf{90.80 \pm 0.058}$ | $\mathbf{94.24 \pm 0.32}$ | $\mathbf{94.18 \pm 0.21}$ | $\mathbf{93.80 \pm 0.2}$ | $\mathbf{53.21 \pm 0.65}$ |

| Sample | Approach | | | |
|---|---|---|---|---|
| | EDL | AEDL | DRO | AREO |
| Bird1 | 137 | 1533 | 3109 | 4 |
| Bird2 | 4548 | 4225 | 3985 | 100 |
| Bird3 | 1928 | 2258 | 4452 | 183 |
| Boat | 1274 | 1520 | 1249 | 223 |
| Truck | 3308 | 1318 | 4208 | 51 |

(a) Representative difficult samples        (b) Ranking of samples

Figure 2: (a) Top row: minority class; bottom-row: majority classes; (b) sample ranking.

distribution, leads to a rather poor OSD performance due to the reasons as analyzed in prior sections. Similarly, AEDL does not perform better than the standard EDL due to the lack of fine-tuning of the difficult examples from the majority classes that become inseparable from the open set samples with a high predicted uncertainty score. Table 10 in the Appendix also shows the closed set performance as a reference. Further, for the deeper insight on the superior OSD of AREO please refer to Appendix.

## 4.4 QUALITATIVE EXAMPLES

We perform a qualitative analysis to further assess the effectiveness of AREO. Figure 2 (a) top row shows representative testing samples from the minority class ('bird') in Cifar10. These images appear to be difficult even for the humans to identify the bird as only a small part is visible. Thus, EDL, AEDL, and DRO assign a relative higher uncertainty score for them. As a result, many open set samples may be assigned a relatively lower uncertainty score, leading to false negative detection on these samples. Figure 2 (b) shows the ranking of these samples according to the uncertainty scores (a lower ranking indicates a lower uncertainty). In contrast, AREO assigns much lower rankings for these birds objects. This analysis justifies the effectiveness of AREO for detecting minority class data samples in the closed set. Similarly, Figure 2 (a) bottom row show representative images from some majority classes. Again, AREO is able to recognize these difficult samples and assign a relatively low uncertainty score to avoid them being mis-identified as open set samples as shown by Figure 2 (b).

## 5 CONCLUSION

In this paper, we focus on open set detection from imbalanced closed set data. To address the fundamental challenge due to the interplay between the minority-class samples and difficult samples from the majority classes, we propose an important extension of DRO to the evidential learning setting, leading to a novel Adaptive Robust Evidential Optimization (AREO) model. As an evidential learning model, AREO effectively breaks the closed set assumption by explicitly modeling the uncertainty mass that is uniquely suitable for detecting open set samples. An adaptive DRO training process is achieved through multi-scheduler learning to achieve an optimal training behavior. The experimentation conducted on five real-world datasets with diverse types of open set data samples justifies the effectiveness of the proposed model.

## ACKNOWLEDGEMENT

This research was supported in part by an NSF IIS award IIS-1814450 and an ONR award N00014-18-1-2875. The views and conclusions contained in this paper are those of the authors and should not be interpreted as representing any funding agency.

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

# Appendix

In this appendix, we first summarize the major notations used in the main paper in Appendix A. We then present the detailed description of the training process obtained through a bi-level optimization in Appendix B. Proofs of main theoretical results are provided in Appendix C. Finally, we present more detailed experimental dataset, result, and settings in Appendix D. The link to the source code is provided in Appendix E.

## A SUMMARY OF NOTATIONS

Table 2 summarizes all the major symbols along with their descriptions.

Table 2: Symbols with Descriptions

| Notation | Description |
|---|---|
| $b_c$ | Belief mass associated with class $c$ |
| $C$ | Total number of classes |
| $e_c$ | Evidence for the $c^{th}$ singleton |
| $u$ | Uncertainty mass |
| $\alpha$ | Dirichlet Parameters |
| $p_c$ | Probability for the $c^{th}$ singleton |
| $\mathbf{y}_n$ | One hot encoded $C$ dimensional multinomial variable |
| $y_{nc}$ | Class label for the $n^{th}$ data sample for class $c$ |
| $p_{nc}$ | Probability of the $n^{th}$ data sample belonging to class $c$ |
| $\eta_t$ | Uncertainty set size for AREO |
| $\beta$ | Hyperparameter controlling the schedule |
| $\gamma$ | Hyperparameter controlling the emphasis in a minority class |
| $\mathbf{p}$ | Probability distribution in the DRO framework |
| $\mathcal{P}^{DRO}$ | Uncertainty set of DRO |
| $\mathcal{P}^{ARO}$ | Uncertainty set of ARO |
| $\Theta$ | Evidential network parameters |
| $l_n^{EL}(\Theta)$ | Evidential loss with the $n^{th}$ data sample |
| $\mathcal{L}^{AREL}(\Theta)$ | Adaptive robust evidential Loss |
| $\mathcal{F}$ | Set of Different classifiers |
| $\sigma_k$ | Weight associated with $k^{th}$ weak learner |
| p(c) | Weight associated with the $c^{th}$ class from Eq. (6) |
| $\widetilde{p(c)}$ | Readjusted weight associated with the $c^{th}$ class from Eq. (11) |
| $\mathbf{w}$ | Mixing weights associated with the `MSF` to control uncertainty set $\eta_t$ |
| $\mathbf{w}'$ | Mixing weights associated with `MSF` to readjust the class-specific weights |
| $\boldsymbol{\beta}$ | Set of Specific parameters for the SFs to control uncertainty set $\eta_t$ |
| $\boldsymbol{\beta}'$ | Set of Specific parameters for the SFs to readjust the class-specific weights |
| $\mathbf{W}$ | `MSF` Parameter sets associated in our model training |
| $T$ | Total number of Epcochs |

## B TRAINING THROUGH BI-LEVEL OPTIMIZATION

Our training involves a bi-level optimization, where we jointly optimize the network parameter $\Theta$ along with the `MSF` parameters $\mathbf{W}$. Algorithm 1 shows the overall training process based on the population based optimization. We randomly initialize the `MSF` parameters $\mathbf{W}_p$ and network parameters $\Theta_p$ from the corresponding spaces $\mathcal{H}$ and $\Theta_{param}$ respectively shown in Line 3. We perform this initialization for $P$ different models. Next, in each epoch we independently optimize $P$ models using the proposed objective function defined in Eq. (6). After $s$ epochs, we evaluate the accuracy of each model by using 'eval' as the evaluation metric in the validation set. It should be noted that in our case, we used closed set classification performance (MAP) as 'eval' metric. We identify $\widehat{P}$ (with $\widehat{P} < P$) worst performing models and replace their model parameters by the randomly selected model parameters from set of $b$ highest accurate models. This process is known as

exploitation and is demonstrated in Line 12. MSF parameters for those worst performing model can be obtained either through random selection from the original space $\mathcal{H}$ or through small perturbation of the $\mathbf{W}$ of the model whose parameter is copied. This process is called exploration as we are searching for the new MSF parameters and is shown in Line 13. The best performing model parameters and accuracy are stored in the $\Theta^*$ and $acc^*$ respectively. Finally, the best model $\Theta^*$ is returned as the optimal model for the testing.

---

**Algorithm 1: Multi-Scheduler Learning Process**

---

**Input:** $\mathcal{H}$, $P$, $s$, eval, $\widehat{P}$, $T$
1  **Initialize:** epoch = 0, $\Theta^*$ = None, $acc^* = None$
2  **for** $p \in [P]$ **do**
3     $\Theta_p, \mathbf{W}_p \leftarrow$ initialize $(\Theta_{param}, \mathcal{H})$
4  **while** *epoch<T* **do**
5     $\Theta_p \leftarrow$ optimize$(\Theta_p | \mathbf{W}_p), p \in [P]$
6     **if** *epoch%s = 0* **then**
7         $acc_p \rightarrow eval(\Theta_p, \mathbf{W}_p), p \in [P]$
8         $sorted\_idx \leftarrow \arg \text{sortDesc}\{acc_p\}_{p=1}^P$
9         $bottom\_idx \leftarrow sorted\_idx[: -\widehat{P}]$
10        $top\_idx \leftarrow sorted\_idx[: \widehat{P}]$
11       **for** $idx \in bottom\_idx$ **do**
12          $\Theta\_idx, j \leftarrow$ uniform$(\{\Theta_j\}_j^{top\_idx})$
13          $\mathbf{W}\_idx \leftarrow explore(\mathcal{H}, \mathbf{W}_j)$
14       $best\_model\_idx \leftarrow top\_idx[0]$
15       **if** $\Theta^*$ *not None* **then**
16         **if** $acc^* < acc_{best\_model\_idx}$ **then**
17           $acc^* = acc_{best\_model\_idx}$
18           $\Theta^* = \Theta_{best\_model\_idx}$
19       **else**
20         $\Theta^* = \Theta_{best\_model\_idx}$
21     $epoch \leftarrow epoch + 1$
22  return $\Theta^*$ with the highest acc

---

The optimization specified in (9) involves an inequality constraint, which incurs a higher computational overhead. Therefore, in our actual optimization process, we consider a regularized version of the AREO loss as follows:

$$\mathcal{L}^{\text{AREL}} = \max_{\mathbf{p} \geq 0, \mathbf{p}^\top \mathbb{1} = 1} \sum_{n=1}^N p_n l_n^t - \lambda D_f \left( \mathbf{p} \| \frac{\mathbb{1}}{N} \right) \tag{18}$$

Solving the above maximization problem leads to a closed form solution for $\mathbf{p}^*$ as shown by the following lemma. It should be noted that the role of the $\lambda$ is exactly opposite as that of the $\eta$. Specifically, we start from a high $\lambda$ so that the model gives equal emphasis to all data samples. Next, in each step we decrease $\lambda$ using the following Equation

$$\lambda_t = \lambda_{t-1} \text{MSF}(\mathbf{w}, \boldsymbol{\beta}, t, T) \tag{19}$$

Decreasing $\lambda$ helps the model focus on the difficult samples as training progresses.

**Lemma 4.** *Assuming that $D_f$ is the KL divergence, then solving* (18) *leads to the following solution*

$$\mathcal{L}^{AREL} = \sum_{n=1}^N p_n^* l_n^t \tag{20}$$

*where $p_n^*$ is given by*

$$p_n^* = \frac{\exp\left(\frac{l_n^t}{\lambda}\right)}{\sum_{j=1}^N \exp\left(\frac{l_j^t}{\lambda}\right)} \tag{21}$$

The detailed proof is below in C.

## C    PROOFS OF THEORETICAL RESULTS

In this section, we present the detailed proofs for Lemmas 1, 2, 4, and Theorem 3.

**Proof of Lemma 1:**    By setting $\eta \to 0$, we have $D_f(\mathbf{p} \| \frac{1}{N}) \to 0$. This implies that $\mathbf{p}$ is uniform with each element as $\frac{1}{N}$. As a result, the optimization problem becomes

$$\mathcal{L}^{DRO}(\Theta) = \frac{1}{N} \sum_{n=1}^{N} l_n^{EL}(\Theta) \tag{22}$$

**Proof of Lemma 2:**    With $\eta \to \infty$, the uncertainty set defined in (5) reduces to the following

$$\mathcal{P}^{DRO} := \left\{ \mathbf{p} \in \mathbb{R}^N : \mathbf{p}^\top \mathbb{1} = 1, \mathbf{p} \geq 0 \right\} \tag{23}$$

Now, the corresponding Lagrangian form of (6) becomes

$$\mathcal{L}^{DRO}(\Theta, \mathbf{u}, \lambda) = \sum_{n=1}^{N} \left( p_n l_n^{EL}(\Theta) + u_n p_n \right) + \mu \left( \sum_{n=1}^{N} p_n - 1 \right) \tag{24}$$

where $u_n$ and $\mu$ are Lagrangian multipliers. Taking gradient with respect to $p_n$ and setting it zero, we get

$$l_n^{EL}(\Theta) + u_n + \mu = 0 \tag{25}$$

Let $k = \arg\max_n l_n^{EL}(\Theta)$ be the index of data sample with the maximum loss (and assuming it is unique). Then, the following holds true

$$u_k < u_n; \quad \forall n \in [1, N], n \neq k \tag{26}$$

This consequently leads to $u_n > 0, \forall n \in [1, N], n \neq k$. Due to the KKT conditions,

$$u_n p_n = 0; \quad \forall n \in [1, N] \tag{27}$$

we have $p_n = 0, \forall n \in [1, N], n \neq k$. By using the following constraint

$$\sum_{n=1}^{N} p_n = 1 \tag{28}$$

we have the following conclusion

$$p_n = \begin{cases} 1, \text{if } n = k \\ 0, \text{otherwise} \end{cases} \tag{29}$$

This means our optimization reduces to the following

$$\mathcal{L}^{AREL}(\Theta) = \max_n l_n^{EL}(\Theta) \tag{30}$$

which proves the lemma.

**Proof of Lemma 4:**    The the Lagrangian of the regularized loss in (18) is

$$\mathcal{L}^{AREL}(\Theta, v, \lambda) = \sum_{n=1}^{N} p_n l_n^t - \lambda \left( \sum_{n=1}^{N} p_n \log p_n + \log N \right) + v \left[ \left( \sum_{n=1}^{N} p_n \right) - 1 \right] \tag{31}$$

where $v$ is the Lagrangian multiplier. Taking the derivative with respect to $p_n$ and setting it to 0:

$$l_n^t - \lambda \log p_n - \lambda + v = 0 \tag{32}$$

Simplifying above equation, we get $p_n$ as

$$p_n = \exp \left( \frac{l_n^t + v}{\lambda} - 1 \right) \tag{33}$$

Using the summation constraint over $p_n$ *i.e.,* $\sum_{n=1}^{N} p_n = 1$, it leads to following

$$\sum_{n=1}^{N} \exp\left(\frac{l_n^t + v}{\lambda} - 1\right) = 1 \tag{34}$$

Solving the above equation we get the expression for $v$ as follows

$$v = \lambda \log\left(\frac{1}{\sum_{n=1}^{N} \exp\left(\frac{l_n^t}{\lambda} - 1\right)}\right) \tag{35}$$

Substituting the $v$ value into (33) gives

$$p_n = \frac{\exp\left(\frac{l_n^t}{\lambda}\right)}{\sum_{n=1}^{N} \exp\left(\frac{l_n^t}{\lambda}\right)} \tag{36}$$

The concludes the proof of Lemma 4.

**Proof of Theorem 3.** AdaBoost can be achieved through alternative optimization between a classification function $f$ and the worst case probability solution (Freund & Schapire, 1997). To show equivalence with the proposed AREO, our proof includes three steps: (i) a specially designed deep neural network (DNN) architecture and a loss function adapted to match the learning process of AdaBoost, (ii) projected functional sub-gradient descent to optimize the classification function $f$, and (iii) optimizing the worst case probability solution.

*Step 1: A specially designed DNN.* Let $\phi(\mathbf{x}) \in \mathcal{R}^M$ denote a $M$-dimensional feature vector learned using a DNN. By applying a fully connected linear layer with a weight matrix $W \in \mathbb{R}^{K \times M}$ on top of the feature vector, we obtain a set of $K$ (discriminative) functions: $\mathbf{f} = (f_1, ..., f_K)^\top = W\phi(\mathbf{x})$. Then, the final output of the DNN is obtained by aggregating these $K$ functions, leading to $f = \boldsymbol{\sigma}^\top \mathbf{f}$, where $\boldsymbol{\sigma} = (\sigma_1, ..., \sigma_K)^\top$. As a result of this design, the final function output by the DNN can be regarded as lying in the linear span of a set of functions $\mathcal{F} = \{f_1, ..., f_K\}$, given by

$$LS(\mathcal{F}) = \left\{f : f = \sum_{k=1}^{K} \sigma_k f_k, 1 \leq k \leq K, \sigma_k \in (-\infty, \infty)\right\} \tag{37}$$

Training of AREO involves alternating between re-weighting using the worst case probability distribution and updating the prediction function $f$. Next, we prove that given the specially designed DNN, we can exactly optimize the classification function $f$ by keeping the worst case probability fixed and vice versa.

*Step 2: Optimizing the classification function $f$ under the worst case probability.* We first formulate the distributional robust evidential loss, which is given by

$$\mathcal{L}^{AREL} = \max_{\mathbf{p} \in \mathcal{P}^{ARO}} \sum_{n=1}^{N} p_n \mathcal{L}_n(f) \tag{38}$$

where $\mathcal{L}_n(f)$ is the loss associated with the datasample $\mathbf{x}_n$. Then, the optimal $f^*$ can be obtained by minimizing the distributional robust loss:

$$f^* = \min_{f \in LS(\mathcal{F})} \mathcal{L}^{AREL} \tag{39}$$

This optimization involves a nonconvex loss $\mathcal{L}^{AREL}$. To ensure the convergence of $f$ to a stationary point, we adapt the ProbAbilistic Gradient Estimator (PAGE) technique (Li et al., 2021) to the DRO setting (shown in Algorithm 2) which ensures the convergence in $\mathcal{O}(b + \frac{b}{\epsilon^2})$ steps with $b$ being the batch size. Please refer to Theorem 6 further details.

To show that an optimal $f^*$ can be achieved, we first verify that the specially designed DNN and the loss function as described above meet a number key conditions as specified by (Blanchet et al., 2019): (i) the loss functional $\mathcal{L}^{AREL}$ is $L$-smooth, (ii) for two different functions $f^1, f^2 \in LS(\mathcal{F})$, $f^1(\phi(\mathbf{x}_n)) \neq f^2(\phi(\mathbf{x}_n))$, and (iii) $LS(\mathcal{F})$ has a finite dimensional basis. First, (i) is true

because $\mathcal{L}^{AREL}$ is the convex combination of the losses $\mathcal{L}_n(f)$. As each individual loss involves the ReLU term with ReLU added in the output of DNN (to ensure non-negativity of the evidence), the resulting convex combination may not be smooth. Therefore, we use the SoftPlus which is smooth function to approximate the ReLU. The the convex combination of SoftPlus results in the function $\mathcal{L}^{AREL}$ to be $L$-smooth. Second, the rich and high dimensional input data (*i.e.,* diverse images) and the feature encoding through the deep architecture of the DNN ensures (ii) is true. Last, since the dimensionality of the weight matrix $W$ is $K \times M$, it implies that the dimensionality of the basis of $LS(\mathcal{F})$ is bounded by $K$, so (iii) holds true.

The smoothness of $\mathcal{L}^{AREL}$ ensures that a stationary solution is achieved within the $\mathbb{O}(b + \frac{b}{\epsilon^2})$ gradient steps. This allows us to have a guaranteed stationary solution with $\mathbb{E}[\|\nabla \mathcal{L}^{AREL}\|] \leq \epsilon$ in a non-convex optimization setting. Furthermore, since $\mathcal{L}^{AREL}$ is a functional on $f$, the next two conditions ensure that the functional gradient exists and can be evaluated (Blanchet et al., 2019). During the optimization process, we need to make sure that the trajectory of the functional gradient lies in the space $LS(\mathcal{F})$, which can be achieved through functional gradient projection.

*Step 3: Optimizing the worst case probability solution.* Let $f_t$ denote the optimal classification function for the current iteration $t$. Next, we continue to optimize the worst case probability solution. The following lemma shows that such an optimal solution exists.

**Lemma 5.** *Assuming that $\mathcal{L}_n(f_t)$ has a finite exponential moment with $\alpha \geq 0$ being sufficiently large and*

$$\eta_t = \beta^* \psi^{'}(\beta^*) - \psi(\beta^*) \tag{40}$$

*the worst case probability is given as*

$$p_n^* = \frac{\exp\left(\frac{\mathcal{L}_n(f_t)}{\alpha}\right)}{\sum_{j=1}^N \exp\left(\frac{\mathcal{L}_j(f_t)}{\alpha}\right)} \tag{41}$$

*where $\beta^* = \frac{1}{\alpha^*}$, $\alpha^* \geq 0$ be the optimal $\alpha$, and $\psi(\beta) = \log\left[\frac{\sum_{n=1}^N \exp(\beta \mathcal{L}_n(f_t))}{N}\right]$.*

*Proof.* Taking the derivative of the Lagrangian for the optimization problem given in (15) with respect to $p_n$ leads to

$$\mathcal{L}_n(f_t) - \alpha \log p_n - \alpha + u_n = 0 \tag{42}$$

where $u_n$ is the Lagrangian multiplier for the constraint $\mathbf{p} \geq 0$ and $\alpha$ is the Lagrange multiplier for the DRO constraint with the size defined by $\eta_t$. Simplification of the above expression yields

$$\log p_n = \frac{\mathcal{L}_n(f_t)}{\alpha} + \frac{u_n - \alpha}{\alpha} \tag{43}$$

For some $\lambda^{'}$ with $p_n = \lambda^{'} \exp\left(\frac{\mathcal{L}_n(f_t)}{\alpha}\right)$, a candidate solution is

$$p_n^* = \frac{\exp\left(\frac{\mathcal{L}_n(f_t)}{\alpha}\right))}{\sum_{j=1}^N \exp\left(\frac{\mathcal{L}_j(f_t)}{\alpha}\right)} \tag{44}$$

The above equation involves the expression in terms of the Lagrangian multiplier. By leveraging the sufficiency result presented in Chapter 8 Theorem 1 of (Luenberger, 1997), we can find the relationship between the multiplier and our constraint parameter $\eta_t$. As such, our optimal solution can be expressed in terms of original constraint. Suppose that we can find $\alpha^* \geq 0$ and $\mathbf{p}^* \in \mathcal{P}^{ARO}$ such that $\mathbf{p}^*$ maximizes (15) for $\alpha = \alpha^*$ and $\sum_{n=1}^N p_n^* \log p_n^* = \eta_t$ with the optimal solution defined in (44). Considering this, we have the following

$$\eta_t = \sum_{n=1}^N p_n^* \log p_n^* = \sum_{n=1}^N p_n^* \frac{\mathcal{L}_n(f_t)}{\alpha^*} - \log\left(\sum_{j=1}^N \exp\left(\frac{\mathcal{L}_j(f_t)}{\alpha^*}\right)\right) = \beta^* \psi^{'}(\beta^*) - \psi(\beta^*) \tag{45}$$

where we define $\beta^* = \frac{1}{\alpha^*}$ and $\psi(\beta) = \log \sum_{n=1}^N \exp\left(\beta \mathcal{L}(f_t)\right)$. This allows us to express the Lagrangian multiplier using $\eta_t$. Next, we verify that there exists an unique solution defined in (44) by

leveraging the convexity of the exponential function. Specifically, substituting (44) in (15), we get the following

$$\sum_{n=1}^{N} p_n^* \mathcal{L}_n(f_t) - \alpha \left( \sum_{n=1}^{N} p_n^* \log p_n^* \right) = \alpha \log \sum_{n=1}^{N} \exp \left( \frac{\mathcal{L}_n(f_t)}{\alpha} \right) \tag{46}$$

If we could show the following inequality holds true

$$\alpha \log \sum_{n=1}^{N} \exp \left( \frac{\mathcal{L}_n(f_t)}{\alpha} \right) \geq \sum_{n=1}^{N} p_n \mathcal{L}_n(f_t) - \alpha \sum_{n=1}^{N} p_n \log p_n \tag{47}$$

then we can claim that the above candidate solution is the optimal solution. Rearranging the terms, we get the following

$$\sum_{n=1}^{N} \exp \left( \frac{L_n(f_t)}{\alpha} \right) \geq \exp \sum_{n=1}^{N} \left( \frac{p_n \mathcal{L}_n(f_t)}{\alpha} - p_n \log p_n \right) \tag{48}$$

This can be shown as

$$\sum_{n=1}^{N} \exp \left( \frac{\mathcal{L}_n(f_t)}{\alpha} \right) = \sum_{n=1}^{N} p_n p_n^{-1} \exp \left( \frac{\mathcal{L}_n(f_t)}{\alpha} \right) = \sum_{n=1}^{N} p_n \exp \left( \frac{\mathcal{L}_n(f_t)}{\alpha} - \log p_n \right)$$

Now applying Jensen inequality to the exponential function $\psi \left( \frac{\sum x_i}{n} \right) \leq \frac{\sum \psi(x_i)}{n}$, we have the following

$$\sum_{n=1}^{N} p_n \exp \left( \frac{\mathcal{L}_n(f_t)}{\alpha} - \log p_n \right) \geq \exp \left( \sum_{n=1}^{N} \frac{p_n \mathcal{L}_n(f_t)}{\alpha} - p_n \log p_n \right) \tag{49}$$

This completes the proof of the lemma. $\qquad \square$

**Theorem 6.** *Suppose that $\mathcal{L}^{AREL}$ holds the L-smoothness criteria with following inequality*

$$\|\nabla \mathcal{L}^{AREL}(f^1) - \nabla \mathcal{L}^{AREL}(f^2)\| \leq L\|f^1 - f^2\| \tag{50}$$

*Then choosing a learning rate $\gamma \leq \frac{1}{L \left( 1 + \sqrt{\frac{1-p}{pb'}} \right)}$ with minibatch size $b = n$, secondary minimbatch size $b' < b$, the number of iterations required to be performed by our algorithm for finding $\epsilon$-approximate solution* i.e., $\mathbb{E}[\|\nabla \mathcal{L}^{AREL}(\hat{f}_T)\| \leq \epsilon]$ *can be bounded by the following:*

$$T = \frac{2\Delta_0 L}{\epsilon^2} \left( 1 + \sqrt{\frac{1-p}{pb'}} \right) \tag{51}$$

*Further the gradient complexity in terms of number of gradient steps is given as*

$$N_{grad} = b + \frac{2\Delta_0 L}{\epsilon^2} \left( 1 + \sqrt{\frac{1-p}{pb'}} \right) (pb + (1-p)b') \tag{52}$$

Before giving the formal proof, we first show two lemmas that are used during the proof.

**Lemma 7.** *The L-smoothness condition given by Eq.* (50)*, leads to the following inequality*

$$\mathcal{L}^{AREL}(f^2) \leq \mathcal{L}^{AREL}(f^1) + \langle \nabla \mathcal{L}^{AREL}(f^1), f^2 - f^1 \rangle + \frac{L}{2}\|f^2 - f^1\|^2, \ \forall f^1, f^2 \in \mathcal{R}^m. \tag{53}$$

*where $\langle a, b \rangle = a^T b$, and $\| \cdot \|$ is the Euclidean norm.*

**Proof of Lemma 7.**   For the completeness the proof of the above Lemma is as follow.

$$\mathcal{L}^{AREL}(f^2)$$
$$\leq \mathcal{L}^{AREL}(f^1) + \int_0^1 \langle \nabla \mathcal{L}^{AREL}(f^1) + \tau(f^2 - f^1)), f^2 - f^1 \rangle d\tau$$
$$= \mathcal{L}^{AREL}(f^1) + \langle \nabla \mathcal{L}^{AREL}(f^1), f^2 - f^1 \rangle$$
$$+ \int_0^1 \langle \nabla \mathcal{L}^{AREL}(f^1 + \tau(f^2 - f^1)) - \nabla \mathcal{L}^{AREL}(f^2), f^2 - f^1 \rangle d\tau$$

Cauchy-Schwarz inequality $\langle u, v \rangle \leq \|u\|\|v\|$ leads to the following

$$\mathcal{L}^{AREL}(f^2)$$
$$\leq \mathcal{L}^{AREL}(f^1) + \langle \nabla \mathcal{L}^{AREL}(f^1), f^2 - f^1 \rangle)$$
$$+ \int_0^1 \|\nabla \mathcal{L}^{AREL}(f^1 + \tau(f^2 - f^1)) - \nabla \mathcal{L}^{AREL}(f^1)\|\|f^2 - f^1\| d\tau$$

Now lets use the L-smoothness assumption from Eq. (50), we have

$$\mathcal{L}^{AREL}(f^2)$$
$$\leq \mathcal{L}^{AREL}(f^1) + \langle \nabla \mathcal{L}^{AREL}(f^1), f^2 - f^1 \rangle) + \int_0^1 L\tau \|f^2 - f^1\|^2 d\tau$$
$$= \mathcal{L}^{AREL}(f^1) + \langle \nabla \mathcal{L}^{AREL}(f^1), f^2 - f^1 \rangle) + \frac{L}{2} \|f^2 - f^1\|^2$$

Now, we provide another important Lemma required to prove the above Theorem based on Lemma 7

**Lemma 8.** *Considering L-smoothness assumption in Eq. (50), and let* $f_{t+1} := f_t - \gamma g_t$. *Then for any* $g_t \in \mathcal{R}^M$ *and* $\gamma > 0$ *we have the following*

$$\mathcal{L}^{AREL}(f_{t+1})$$
$$\leq \mathcal{L}^{AREL}(f_t) - \frac{\gamma}{2}\|\nabla \mathcal{L}^{AREL}(f_t)\|^2 - \left(\frac{1}{2\gamma} - \frac{L}{2}\right)\|f_{t+1} - f_t\|^2 + \frac{\gamma}{2}\|g_t - \nabla \mathcal{L}^{AREL}(f_t)\|^2$$

$$(54)$$

**Proof of Lemma 8.** *Let $\bar{f}_{t+1} := f_t - \gamma \nabla \mathcal{L}^{AREL}(f_t)$. Then using L-smoothness of $\mathcal{L}^{AREL}$, we have the following*

$$
\mathcal{L}^{AREL}(f_{t+1})
$$
$$
\leq \mathcal{L}^{AREL}(f_t) + \langle \nabla \mathcal{L}^{AREL}(f_t), f_{t+1} - f_t \rangle + \frac{L}{2}\|f_{t+1} - f_t\|^2
$$
$$
= \mathcal{L}^{AREL}(f_t) + \langle \nabla \mathcal{L}^{AREL}(f_t) - g_t, f_{t+1} - f_t \rangle + \langle g_t, f_{t+1} - f_t \rangle + \frac{L}{2}\|f_{t+1} - f_t\|^2
$$
$$
= \mathcal{L}^{AREL}(f_t) + \langle \nabla \mathcal{L}^{AREL}(f_t) - g_t, -\gamma g_t \rangle - \left(\frac{1}{\gamma} - \frac{L}{2}\right)\|f_{t+1} - f_t\|^2
$$
$$
= \mathcal{L}^{AREL}(f_t) + \gamma\|\nabla \mathcal{L}^{AREL}(f_t) - g_t\|^2 - \gamma\langle \nabla \mathcal{L}^{AREL}(f_t) - g_t, \nabla \mathcal{L}^{AREL}(f_t) \rangle
$$
$$
- \left(\frac{1}{\gamma} - \frac{L}{2}\right)\|f_{t+1} - f_t\|^2
$$
$$
= \mathcal{L}^{AREL}(f_t) + \gamma\|\nabla \mathcal{L}^{AREL}(f_t) - g_t\|^2 - \frac{1}{\gamma}\langle f_{t+1} - \bar{f}_{t+1}, f_t - \bar{f}_{t+1} \rangle
$$
$$
- \left(\frac{1}{\gamma} - \frac{L}{2}\right)\|f_{t+1} - f_t\|^2
$$
$$
= \mathcal{L}^{AREL}(f_t) + \gamma\|\nabla \mathcal{L}^{AREL}(f_t) - g_t\|^2 - \left(\frac{1}{\gamma} - \frac{L}{2}\right)\|f_{t+1} - f_t\|^2
$$
$$
- \frac{1}{2\gamma}\left(\|f_{t+1} - \bar{f}_{t+1}\|^2 + \|f_t - \bar{f}_{t+1}\|^2 - \|f_{t+1} - f_t\|^2\right)
$$
$$
= \mathcal{L}^{AREL}(f_t) + \gamma\|\nabla \mathcal{L}^{AREL}(f_t) - g_t\|^2 - \left(\frac{1}{\gamma} - \frac{L}{2}\right)\|f_{t+1} - f_t\|^2
$$
$$
- \frac{1}{2\gamma}\left(\|\gamma^2\|\nabla \mathcal{L}^{AREL}(f_t) - g_t\|^2 + \gamma^2\|\nabla \mathcal{L}^{AREL}(f_t)\|^2 - \|f_{t+1} - f_t\|^2\right)
$$
$$
= \mathcal{L}^{AREL}(f_t) - \frac{\gamma}{2}\|\nabla \mathcal{L}^{AREL}(f_t)\|^2 - \left(\frac{1}{2\gamma} - \frac{L}{2}\right)\|f_{t+1} - f_t\|^2 + \frac{\gamma}{2}\|g_t - \nabla \mathcal{L}^{AREL}(f_t)\|^2
$$

This completes the Proof of Lemma 8. The last term is the variance and it can be bounded using the following lemma.

**Lemma 9.** *Suppose that the smoothness assumption in Eq. (50) holds. If the gradient estimator $g_{t+1}$ is defined in Algorithm 2 Line 13, then we have the following*

$$
\mathbb{E}[\|g_{t+1} - \nabla \mathcal{L}^{AREL}(f_{t+1})\|^2] \leq (1 - p_t)\|g_t - \nabla \mathcal{L}^{AREL}(f_t)\|^2 + \frac{(1 - p_t)L^2}{b'}\|f_{t+1} - f_t\|^2 \quad (55)
$$

**Proof of Lemma 9.** *According to Algorithm 2, we have the following*

$$
g_{t+1} = \begin{cases} \frac{1}{b}\sum_{n \in B} a_n(f_{t+1})\nabla\mathcal{L}_n(f_{t+1}) \text{ with probability } p_t \\ g_t + \frac{1}{b'}\sum_{n \in B'}(a_n(f_{t+1})\nabla\mathcal{L}_n(f_{t+1}) - a_n(f_t)\nabla\mathcal{L}_n(f_t)), \quad \text{with probability } 1 - p_t \end{cases}
$$
$$
(56)
$$

*Using this the left hand side of the above lemma can be written as*

$$\mathbb{E}[\|g_{t+1} - \nabla \mathcal{L}^{AREL}(f_{t+1})\|^2]$$

$$= p_t \mathbb{E}\left[\|\frac{1}{b}\sum_{n\in B} a_n(f_{t+1})\nabla\mathcal{L}_n(f_{t+1}) - \nabla\mathcal{L}^{AREL}(f_{t+1})\|^2\right]$$

$$+ (1-p_t)\mathbb{E}\left[\|g_t + \frac{1}{b'}\sum_{n\in B'}(a_n(f_{t+1})\nabla\mathcal{L}_n(f_{t+1}) - a_n(f_t)\nabla\mathcal{L}_n(f_t)) - \nabla\mathcal{L}^{AREL}(f_{t+1})\|^2\right]$$

$$= (1-p_t)\mathbb{E}\left[\|g_t + \frac{1}{b'}\sum_{n\in B'}(a_n(f_{t+1})\nabla\mathcal{L}_n(f_{t+1}) - a_n(f_t)\nabla\mathcal{L}_n(f_t)) - \nabla\mathcal{L}^{AREL}(f_{t+1})\|^2\right]$$

$$= (1-p_t)\mathbb{E}\left[\|g_t - \nabla\mathcal{L}^{AREL}(f_t) + \frac{1}{b'}\sum_{n\in B'}(a_n(f_{t+1})\nabla\mathcal{L}_n(f_{t+1}) - a_n(f_t)\nabla\mathcal{L}_n(f_t))\right.$$

$$+ (1-p_t)\mathbb{E}\left[-\nabla\mathcal{L}^{AREL}(f_{t+1}) + \nabla\mathcal{L}^{AREL}(f_t)\|^2\right]$$

$$= (1-p_t)\mathbb{E}\left[\|\frac{1}{b'}\sum_{n\in B'}(a_n(f_{t+1})\nabla\mathcal{L}_n(f_{t+1}) - a_n(f_t)\nabla\mathcal{L}_n(f_t)) - \nabla\mathcal{L}^{AREL}(f_{t+1}) + \nabla\mathcal{L}^{AREL}(f_t)\|^2\right]$$

$$+ (1-p_t)\|g_t - \nabla\mathcal{L}^{AREL}(f_t)\|^2$$

$$= \frac{1-p_t}{b'^2}\mathbb{E}\left[\sum_{n\in B'}\|(a_n(f_{t+1})\nabla\mathcal{L}_n(f_{t+1}) - a_n(f_t)\nabla\mathcal{L}_n(f_t))\right]$$

$$- \frac{1-p_t}{b'^2}\mathbb{E}\left[(\nabla\mathcal{L}^{AREL}(f_{t+1}) - \nabla\mathcal{L}^{AREL}(f_t))\|^2\right] + (1-p_t)\|g_t - \nabla\mathcal{L}^{AREL}(f_t)\|^2$$

$$\leq \frac{(1-p_t)L^2}{b'}\|\mathcal{L}^{AREL}(f_{t+1}) - \mathcal{L}^{AREL}(f_t)\|^2 + (1-p_t)\|g_t - \nabla L^{AREL}(f_t)\|^2$$

*Using the L-smoothness assumption in Eq. (50), we have*

$$\mathbb{E}[\|g_{t+1} - \nabla\mathcal{L}^{AREL}(f_{t+1})\|^2] \leq \frac{(1-p_t)L^2}{b'}\|f_{t+1} - f_t\|^2 + (1-p_t)\|g_t - \nabla\mathcal{L}^{AREL}(f_t)\|^2$$

**Proof of Theorem 6.** We leverage the above lemmas to prove the Theorem. Adding Eq. (54) with $\frac{\gamma}{2p}\times$Eq. (55) and taking expectation results in the following:

$$\mathbb{E}\left[\mathcal{L}^{AREL}(f_{t+1}) - L_*^{AREL} + \frac{\gamma}{2p}\|g_{t+1} - \nabla\mathcal{L}^{AREL}(f_{t+1}\|^2\right]$$

$$\leq \mathbb{E}\left[\mathcal{L}^{AREL}(f_t) - \mathcal{L}_*^{AREL} - \frac{\gamma}{2}\|\nabla\mathcal{L}^{AREL}(f_t)\|^2 - \left(\frac{1}{2\gamma} - \frac{L}{2}\right)\|f_{t+1} - f_t\|^2\right]$$

$$+ \frac{\gamma}{2}\mathbb{E}\left[\|g_t - \nabla\mathcal{L}^{AREL}(f_t)\|^2\right] + \frac{\gamma}{2p}\mathbb{E}\left[(1-p)\|g_t - \nabla\mathcal{L}^{AREL}(f_t)\|^2\right]$$

$$+ \frac{\gamma}{2p}\mathbb{E}\left[\frac{(1-p)L^2}{b'}\|f_{t+1} - f_t\|^2\right]$$

$$= \mathbb{E}\left[\mathcal{L}^{AREL}(f_t) - \mathcal{L}_*^{AREL} + \frac{\gamma}{2p}\|g_t - \nabla\mathcal{L}^{AREL}(f_t)\|^2\right]$$

$$- \mathbb{E}\left[\frac{1}{2\gamma} - \frac{L}{2} - \frac{(1-p)\gamma L^2}{2pb'}\|f_{t+1} - f_t\|^2\right]$$

where $L_*^{AREL}$ is the loss at the optimal $f^*$. Using the inequality of $\frac{1}{2\gamma} - \frac{L}{2} - \frac{(1-p)\eta L^2}{2pb'} \geq 0$, *i.e.,*

$$\gamma \leq \frac{1}{L\left(1 + \sqrt{\frac{1-p}{pb'}}\right)} \tag{57}$$

we can write the following

$$\mathbb{E}[\|g_{t+1} - \nabla\mathcal{L}^{AREL}(f_{t+1})\|^2]$$
$$\leq \mathbb{E}\left[\mathcal{L}^{AREL}(f_t) - \mathcal{L}_*^{AREL} + \frac{\gamma}{2p}\|g_t - \nabla\mathcal{L}^{AREL}(f_t)\|^2 - \frac{\gamma}{2}\|\nabla\mathcal{L}^{AREL}(f_t)\|^2\right]$$

Now let us define $\phi_t := \mathcal{L}^{AREL}(f_t) - \mathcal{L}_*^{AREL} + \frac{\gamma}{2p}\|g_t - \nabla\mathcal{L}^{AREL}(f_t)\|^2$ then we can write the following

$$\mathbb{E}[\phi_{t+1}] \leq \mathbb{E}[\phi_t] - \frac{\gamma}{2}\mathbb{E}[\|\nabla\mathcal{L}^{AREL}(f_t)\|^2] \qquad (58)$$

Now summing from $t = 0$ to $T - 1$ results in the following

$$\mathbb{E}[\phi_T] \leq \mathbb{E}[\phi_0] - \frac{\gamma}{2}\sum_{t=0}^{T-1}\mathbb{E}[\|\nabla\mathcal{L}^{AREL}(f_t)\|^2] \qquad (59)$$

According to the Algorithm 2, $\hat{f}_T$ is chosen from $\{f_t\}_{t\in[T]}$ and $\phi_0 = \mathcal{L}^{AREL}(f_0) - \mathcal{L}_*^{AREL} + \frac{\gamma}{2p}\|g_0 - \nabla\mathcal{L}^{AREL}(f_0)\|^2 = \mathcal{L}^{AREL}(f_0) - \mathcal{L}_*^{AREL} = \Delta_0$, we have

$$\mathbb{E}[\|\nabla\mathcal{L}^{AREL}(\hat{f}_T\|^2] \leq \frac{2\Delta_0}{\gamma T} \qquad (60)$$

Setting $T = \frac{2\Delta_0}{\epsilon^2\gamma}$ and using Jensen's inequality results in the following

$$\mathbb{E}[\|\nabla\mathcal{L}^{AREL}(\hat{f}_T)\|] \leq \mathbb{E}[\|\nabla\mathcal{L}^{AREL}(\hat{f}_T)\|^2] \leq \sqrt{\frac{2\Delta_0}{\gamma T}} = \epsilon \qquad (61)$$

With the following total number of iterations

$$T = \frac{2\Delta}{\epsilon^2\gamma} = \frac{2\Delta_0 L}{\epsilon^2}\left(1 + \sqrt{\frac{1-p}{pb'}}\right) \qquad (62)$$

we can obtain $\epsilon$-approximate stationary point solution. The number of gradient steps required in the Algorithm 2 is given as

$$N_{grad} = b + T(pb + (1-p)b') \qquad (63)$$

Replacing $T$ by Equation (62), we have the following

$$N_{grad} = b + \frac{2\Delta_0 L}{\epsilon^2}\left(1 + \sqrt{\frac{1-p}{pb'}}\right)(pb + (1-p)b') \qquad (64)$$

This proves Theorem 6.

# D  ADDITIONAL EXPERIMENTAL DETAILS

## D.1  DATASET DESCRIPTIOM

**MNIST.** In this dataset, classes corresponding to digits '1', '3', '5', '7', and '9' are treated as closed set classes and the rest as the open set. As the number of data samples per class is not exactly the same, we first sample 5,000 samples per class in the training set. For testing, we sample 1,000 samples per class. To make the dataset imbalanced, we consider class '3' as a minority class and randomly select 30% data samples as compared with other majority classes. The same imbalanced ratio is applied to both training and testing sets. Table 3 shows the number of data samples from both the minority class and majority classes. In addition to the MNIST open set classes as described above, we follow other existing works (Sun et al., 2020) and further test the OSD performance on additional open set samples from three more sources: (1) MNIST-Noise, (2) Noise, and (3) Omnigolot (Lake et al., 2015). More specifically, MNIST-Noise is constructed by adding random noises to the closed set testing samples, Noise consists of random noises, and Omnigolot consists of data samples from the Omnigolot dataset. For those classes, we select the same data samples as those of the closed set samples.

---

**Algorithm 2: Alternative Optimization between f and $p$ using Probabilistic SGD**

---

1 **Initialize:** $f_0$, stepsize $\gamma$, minibatch sizes $b, b' < b$, $p_t \in [0,1]$, t = 0, $p_n(f_0) = 1 \forall n \in [1, b]$

2 **compute** $g_0 = \frac{1}{b} \sum_{n \in B} a_n(f_0) \nabla \mathcal{L}_n^{DRO}(f_0)$ $a_n(f_0) = b * p_n(f_0)$ with $B, B'$ being random minibatch samples with $|B| = b$ and $|B'| = b'$

3 **while** *t<T* **do**

4      $f_{t+1} \leftarrow f_t - \gamma g_t$

5      $prev\_use \sim Ber(p_t)$

6      **if** $prev\_use = 1$ **then**

7          Find loss $\mathcal{L}_n(f)$ associated with datasample $\mathbf{x}_n, \forall n \in B$

8          Find $a_n(f^{t+1}) = b * \frac{\exp\left(\frac{\mathcal{L}_{\setminus}(f_t)}{\alpha}\right)}{\sum_{j=1}^{b} \exp\left(\frac{\mathcal{L}_j(f_t)}{\alpha}\right)}$

9          Find $g_{t+1} = \frac{1}{b} \sum_{n \in B} a_n(f_{t+1}) \nabla \mathcal{L}_n(f^{t+1})$

10      **else**

11          Find loss $\mathcal{L}_n(f)$ associated with data sample $\mathbf{x}_n, \forall n \in B'$

12          Find $a_n(f_{t+1}) = b' * \frac{\exp\left(\frac{\mathcal{L}_{\setminus}(f_t)}{\alpha}\right)}{\sum_{j=1}^{b} \exp\left(\frac{\mathcal{L}_j(f_t)}{\alpha}\right)}$

13          Find $g_{t+1} = g_t + \frac{1}{b'} \sum_{n \in B'} (a_n(f_{t+1}) \nabla \mathcal{L}_n(f_{t+1}) - a_n(f_t) \nabla \mathcal{L}_n(f_t))$

14      $t \leftarrow t + 1$

15 **return** $\hat{f}_T$ chosen from $\{f_t\}_{t \in [T]}$

---

Table 3: Number of closed set samples in different datasets

| Split | Cifar10 | | Cifar100 | | MNIST | | ImageNet | | AHED | |
|---|---|---|---|---|---|---|---|---|---|---|
| | *Majority* | *Minority* | *Majority* | *Minority* | *Majority* | *Minority* | **Majority** | **Minority** | **Majority** | **Minority** |
| Train | 20000 | 1500 | 22500 | 800 | 20000 | 1500 | 2960 | 222 | 5071 | 246 |
| Test | 4000 | 300 | 4500 | 150 | 4000 | 300 | 744 | 56 | 1271 | 64 |

**Cifar10.** In the Cifar10 dataset, classes 'Airplane', 'Automobile', 'Bird', 'Ship', and 'Truck' are considered as the closed set classes and 'Cat', 'Deer', 'Dog', 'Frog', and 'Horse' as open set classes . Furthermore, the class 'Bird' is made as the minority class using the same strategy introduced above. In addition to the open set classes from Cifar10 itself, we further assess the OSD performance with Cifar+10 and Cifar+50. In particular, Cifar+10 includes data samples from the randomly selected 10 classes of the Cifar100 dataset and Cifar+50 includes data samples from 50 randomly selected classes of the Cifar100 dataset.

**Cifar100.** In the Cifar100 dataset, 'living being' related super classes are regarded as the closed set classes and the remaining 'non-living being' related super classes are regarded as the open set classes. We make 'insect' related classes as the minority one.

**ImageNet.** In the ImageNet dataset, we performed experimentation by randomly picking 5 classes as known classes and five classes as unknown classes. Specifically classes 'ant', 'king crab', 'lion', 'French bulldog', 'great white shark' are treated as known classes whereas, classes 'iPod', 'lipstick', 'street sign', 'bookshop', and 'miniskirt' as unknown classes. We make 'king crab' as the minority class in this dataset.

**Architectural Heritage Elements Dataset (AHED).** In this dataset, we pick classes 'bell tower', 'portal', 'gargoyle', 'dome', and 'column' as open set classes whereas, classes 'apse', 'vault', 'altar', 'stained glass', and 'flying buttress' as unknown classes. This is inherently highly imbalanced dataset where number of data points are unevenly distributed across different classes. The class 'portal' is the minority class in this dataset.

### D.2 DETAILED EXPERIMENTAL SETTING

For all datasets, we use LeNet5 (Lecun et al., 1998) as the network architecture with Tanh activation in the CNN layers and SoftPlus activation in the fully connected layers. Training involves the Adam optimizer with a learning rate of $0.001$ and $l2$ regularization coefficient of $0.001$. We initialize the uncertainty set $\lambda_0 = 100$ for all datasets so that model gives the equal emphasis to the all

Table 4: Ablation Study on OSD Results for the Different Datasets.

| Cosine | Exponential | Cifar10 | | | Cifar100 | ImageNet |
|---|---|---|---|---|---|---|
| | | Cifar10 | Cifar+10 | Cifar+50 | | |
| ✓ | | $70.43 \pm 0.48$ | $29.41 \pm 1.86$ | $69.71 \pm 2.32$ | $55.61 \pm 1.85$ | $60.68 \pm 0.79$ |
| | ✓ | $61.62 \pm 0.26$ | $25.17 \pm 1.22$ | $60.94 \pm 1.19$ | $53.22 \pm 0.70$ | $59.59 \pm 1.48$ |
| ✓ | ✓ | $72.48 \pm 4.08$ | $37.14 \pm 2.06$ | $73.87 \pm 1.42$ | $57.52 \pm 1.60$ | $62.02 \pm 1.11$ |
| **Cosine** | **Exponential** | **MNIST** | | | | **AHED** |
| | | MNIST | Noise | MNIST-Noise | Omnigolot | |
| ✓ | | $80.82 \pm 10.49$ | $88.81 \pm 6.42$ | $84.01 \pm 4.30$ | $85.72 \pm 6.55$ | $52.18 \pm 1.18$ |
| | ✓ | $87.38 \pm 2.96$ | $89.85 \pm 7.47$ | $85.29 \pm 6.26$ | $82.44 \pm 5.88$ | $53.08 \pm 1.43$ |
| ✓ | ✓ | $90.80 \pm 0.06$ | $94.25 \pm 0.32$ | $94.18 \pm 0.21$ | $93.80 \pm 0.20$ | $53.21 \pm 0.65$ |

data samples. In terms of MSF parameters ($\mathbf{w}$ and $\mathbf{w}'$) for each model in $P$, we initialize them by uniformly sampling from $[0, 1]$. Next, we follow the training procedure shown in Algorithm 1 with random parameter selection in the exploration phase.

### D.3 ABLATION STUDY ON SCHEDULER FUNCTIONS

Table 4 compares the OSD performance between the MSF obtained through the proposed multi-scheduler learning strategy with a prefixed atomic scheduler function, including cosine and exponential. Due to the lack of expressiveness, a single atomic SF function usually cannot work well for all datasets. For example, in Cifar10, cosine yields a relatively better OSD compared to exponential, whereas in the case of MNIST, exponential produces a relatively higher OSD performance. In contrast, by combining both of them and properly balancing their contribution based upon the nature of the dataset, MSF achieves the best performance in all cases.

Figure 3 below shows the resulting MSF function output for the Cifar10 and Cifar100 datasets that exhibit quite different learning behaviors. For Cifar10, the model can quickly learn from the classes due to relatively easy data samples. As the MSF function decreases quickly, the resulting $\eta$ value in AREO increases quickly making the model focusing mostly on the difficult samples. In case of Cifar100, because of its difficult nature, the model takes more time to learn well from all samples. Only in the latter phase of training, the model starts to put more emphasis on the difficult samples by increasing the $\eta$ value in the AREO.

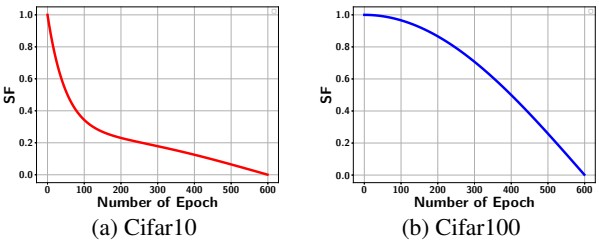

(a) Cifar10        (b) Cifar100

Figure 3: MSF Function output Visualization for Multiple Datasets

### D.4 PERFORMANCE ON MULTIPLE IMBALANCED CLASSES

To demonstrate the capability of handling the multiple minority classes with severe imabalanced condition, we conduct an experiments with multiple minority classes under two different settings discussed below

**Same Cardinality:** In this first setting we demonstrate the ability of our technique handling multiple imbalanced classes. In this case, we consider two minority classes with $c_1 = 6\%$ and $c_2 = 6\%$. In each step, we randomly select minority classes and repeat the experimentation two times and take an average over those runs to get the final score. Table 5 shows the performance for different baselines along with our proposed AREO. As shown, our technique has a far better performance in terms of OSD compared to the existing baselines.

Table 5: OSD (MAP) performance on Multiple Imbalanced Classes with Same Cardinality

| Approach | Cifar10 | | | MNIST | | | |
|----------|---------|----------|----------|-------|-------|-------------|----------|
| | Cifar10 | Cifar+10 | Cifar+50 | MNIST | Noise | MNIST-Noise | Omnigolot |
| EDL | 65.04 | 30.61 | 67.71 | 85.50 | 90.42 | 86.15 | 90.26 |
| AEDL | 58.12 | 23.20 | 58.70 | 87.57 | 94.96 | 89.26 | 90.67 |
| DRO | 53.40 | 22.19 | 58.62 | 60.89 | 29.04 | 37.88 | 33.57 |
| OLTR | 59.06 | 22.06 | 56.85 | 84.24 | 91.65 | 80.26 | 53.68 |
| **AREO** | **71.38** | **31.21** | **68.78** | **90.16** | **98.15** | **93.45** | **95.18** |

**Different Cardinality with Severe Imbalance:** In this setting, we demonstrate the ability of our approach on the multiple imbalanced classes with different cardinality. In this case, we consider two minority classes with $c_1 = 6\%$ and $c_2 = 3\%$. Similar to first case, we randomly select minority classes and repeat the experimentation two times and take an average over those runs to get the final score. Further, this also demonstrates the ability of our technique handling severe type of the imbalanced scenario where $c_2 = 3\%$. Table 6 shows the performance comparison across different techniques. As shown, our AREO has a better performance compared to the existing baselines.

Table 6: OSD (MAP) performance on Multiple Imbalanced Classes with Different Cardinality

| Approach | Cifar10 | | | MNIST | | | |
|----------|---------|----------|----------|-------|-------|-------------|----------|
| | Cifar10 | Cifar+10 | Cifar+50 | MNIST | Noise | MNIST-Noise | Omnigolot |
| EDL | 68.46 | 33.05 | 69.17 | 87.16 | 93.87 | 90.00 | 91.37 |
| AEDL | 59.60 | 23.85 | 62.44 | 87.98 | 92.45 | 89.26 | 89.91 |
| DRO | 57.38 | 22.72 | 61.82 | 67.72 | 80.32 | 66.59 | 71.74 |
| OLTR | 59.33 | 21.71 | 56.00 | 84.46 | 84.94 | 79.94 | 60.28 |
| **AREO** | **70.01** | **33.79** | **71.31** | **89.97** | **96.31** | **94.24** | **92.22** |

## D.5 DETAILED DESCRIPTION OF BASELINES AND ADDITONAL COMPARION RESULT

### D.5.1 DESCRIPTION OF BASELINES

CGDL uses a variational autoencoder that can learn the class conditional posterior distribution for each class in the latent space (Sun et al., 2020). Any sample with a low probability of belonging to any of the classes are regarded as the outliers. CGDL is consistently outperformed by the proposed AREO model. One possible reason is that CGDL may not learn the minority class posterior distribution properly in the latent space. Further, the approach may ignore the difficult samples when forming the posterior distribution. As such, the model is not able to differentiate the minority class and OOD samples as both may have low probability of belonging to any of the classes resulting in the lower performance.

Posterior networks (Charpentier et al., 2020) leverage the normalizing flows to obtain the posterior distribution over the predicted probabilities. In this case, the latent representation is learned using the encoder and per class probability density value for that latent representation is determined to get the posterior distribution. In case of the minority class, the normalizing flow may not learn well to have a high probability density value to the sample. As such, while computing the uncertainty, it may still assign to the low uncertainty to minority class. Further, the model may not learn well to produce the high density value for difficult samples from other classes. As such, it is likely the model may have a confusion between hard samples and OOD samples resulting in difficulty in OSD. As shown in Table 1, the posterior networks have consistently lower performance compared to AREO.

Kopetzki et al. (2021) propose a more robust form of the posterior networks by training the network using randomized smoothing (RS). The key idea is to draw multiple samples $\mathbf{x}_s^i \sim \mathcal{N}(\mathbf{x}^i, \sigma)$ around the input sample $\mathbf{x}^i$. Although this technique has shown improvement over the adversarial attack, it is not designed for the imbalanced situation as demonstrated by its lower OSD performance in Table 1.

We have also included an approach called OLTR proposed by Liu et al. (2019) as a baseline. This work has proposed a way to deal with the OSD in the imbalanced data distribution however it has several limitations. First, the approach is based on the assumption that visual similarity is shared across the minority and majority classes and thereby having robust learning even for the minority class. So, if the minority class is different from other majority classes, the model may not properly

learn the minority classes and thereby the model may be confused between minority and open set samples. Also, the proposed OLTR does not have a mechanism to focus on the difficult samples from the majority classes while training. As such, the model may detect difficult samples as open set samples. Those limitations are reflected in the performance shown in Table 1. Recently, Wang et al. (2022) propose a contrastive loss based approach where minority classes are pushed from the OOD samples in the feature space. Although, this paper also considers the open set detection under class imbalanced-setting, it heavily relies on the selection of open set samples involved during the training process. Table 7 shows the OSD performance with respect to open set training datasets Flower (Nilsback & Zisserman, 2006) and MIT Indoor Scene (Quattoni & Torralba, 2009). As shown, the performance using this method is highly dependent on the selection of the OOD dataset during training process.

Table 7: OSD (MAP) performance on (Wang et al., 2022)

| Openset | Cifar10 | | | Cifar100 |
|---|---|---|---|---|
| | Cifar10 | Cifar+10 | Cifar+50 | |
| Flower | $70.67 \pm 1.69$ | $25.78 \pm 2.08$ | $64.09 \pm 3.00$ | $53.96 \pm 0.57$ |
| Indoor | $72.94 \pm 0.52$ | $31.70 \pm 3.35$ | $69.65 \pm 3.71$ | $55.74 \pm 0.35$ |

### D.5.2 Additional Comparison Results

In this section, we first provide a detailed discussion on recently developed general open set recognition methods, which do not specifically focus on imbalanced data. We then choose some representative methods and present a comparison with the proposed AREO model. Finally, we discuss some recent OSD models designed for few-shot learning under the meta-learning setting.

Chen et al. (2021) conduct Adversarial Reciprocal Point Learning (ARPL), where the adversarial point is generated for each known class in the embedding space by leveraging representations of other known class samples along with the unknown samples generated using an adversarial mechanism. During training, it maximizes the gap between the representations of known class samples and that of the adversarial point. Meanwhile, the model tries to push the unknown samples' representations into a specific region using the adversarial margin constraint. In order to achieve this, diverse and confusing training samples are generated through adversarial learning. Cevikalp et al. (2021) leverage the polyhedral conic function and define two losses. The first loss ensures a good separation among the known classes whereas the second loss achieves the compactness within each class so that the open set samples could be easily rejected. Dhamija et al. (2018) also leverage two losses, where the Entropic open set loss is to ensure that the softmax output for the open set samples are uniformly distributed to the all known classes and the Objectosphere loss aims to assign a higher feature magnitude to the known class samples in the embedding space than those from the open set classes. The proposed approach requires open set datasets to be available during the training set, which may limit its applicability in more general settings. In (Perera et al., 2020), self supervision is performed to construct the decision boundary between classes based on the semantics in the feature space. A generative model is then trained based on the known class samples. Thus, the generated images will be close to that of the closed set class samples. As the open set samples are not seen during the generative modeling process, the produced images will exhibit a high disparity with those of the closed set samples. Finally, Yang et al. (2020) propose the Convolutional Prototype Network (CPN) where a prototype for each known class is constructed in the feature space and two different loss functions are defined. To define the generative loss, generative assumption is followed where the class-specific features are drawn from certain distributions (*e.g.,* Gaussian) with the mean given by the prototype representation. This generative loss helps to reduce the intra-class variance and thereby making the known class sample representation very compact. Thus, the model can reserve more spaces for unknowns, making OSD relatively easier. The second loss (*i.e.,* discriminative loss) encourages the separation of the class samples from different classes based on the distance between sample and prototype representations.

We perform comparison with the first three methods discussed above, including two most recent baselines with competitive OSD performance. We report the comparison results on Cifar 10, Cifar 100, and MNIST datasets in Table 8. As discussed earlier, all these methods are general open set recognition models and hence suffer from a lower OSD performance for the more challenging setting

that involves imbalanced data. The results further justify the effectiveness of the proposed AREL model.

Table 8: OSD (MAP) performance on Multiple Imbalanced Classes with Different Cardinality

| Approach | Cifar10 | | | MNIST | | | | Cifar100 |
|---|---|---|---|---|---|---|---|---|
| | Cifar10 | Cifar+10 | Cifar+50 | MNIST | Noise | MNIST-Noise | Omnigolot | |
| ARPL (Chen et al. (2021)) | 58.90 | 29.33 | 69.00 | 90.27 | 74.57 | 50.38 | 39.08 | 48.74 |
| Agnostophobia (Dhamija et al. (2018)) | 68.50 | 29.11 | 68.36 | 85.03 | 78.43 | 72.72 | 75.76 | 52.93 |
| DC ECPP (Cevikalp et al. (2021)) | 61.22 | 23.54 | 58.30 | 77.56 | 55.80 | 71.91 | 89.44 | 53.22 |
| **AREO** | **72.48** | **37.14** | **73.87** | **90.80** | **94.24** | **94.18** | **93.80** | **57.52** |

In addition to the above baselines, there are also recent OSD models specifically developed for few-shot learning (FSR) under the meta-learning setting. For instance, Jeong et al. (2021) propose a few-shot open-set recognition (FSOSR) model. This approach is designed to work with testing tasks with limited labeled data through meta-learning. The design of the training paradigm is based on episodic learning (Snell et al., 2017) widely used in meta-learning, where query and support sets are constructed by selecting subsets of the meta-training data. In contrast, our model is not designed for few-shot setting through meta-learning. Furthermore, the FSOSR approach in (Jeong et al., 2021) does not consider the imbalanced class distribution, either. Therefore, the problem setting, model training, and evaluation process are all different. Similarly, Liu et al. (2020) propose an oPen set mEta LEaRning (PEELER) algorithm that adapts ProtoNet to FSOSR under the meta-learning setting. There are two key differences between our technique and PEELER. First, PEELER assumes that the unknown samples are also available during the training process. Second, the algorithm is also designed under the meta-learning setting that makes the direct comparison with our approach infeasible.

Kong & Ramanan (2021) propose the OpenGAN model to discriminate the open set samples from the close-set ones. There are some key differences from our work. First, OpenGAN introduces open set samples in the training as well as validation sets whereas our approach does not involve any open set samples in training or validation sets. Second, OpenGAN only works in the binary classification setting where the loss function is proposed to discriminate whether a sample is in the open set or closed set. In this case, the loss function does not perform closed set classification. In contrast, our approach achieves a state-of-the-art OSD performance while ensuring decent closed set performance. Finally, OpenGAN does not have a specific mechanism to handle the imbalanced setting, which is one primary design focus of our approach.

## D.6 Performance Comparison using AUROC

In addition to the MAP scores, the AUROC scores, which are reported in Table 9, also show a consistent trend in terms of OSD performance.

Table 9: OSD (AUC) performance on Multiple Datasets

| Approach | Cifar10 | | | MNIST | | | | Cifar100 |
|---|---|---|---|---|---|---|---|---|
| | Cifar10 | Cifar+10 | Cifar+50 | MNIST | Noise | MNIST-Noise | Omnigolot | |
| EDL | 66.36 | 77.66 | 69.07 | 88.72 | 95.36 | 93.08 | 91.23 | 52.38 |
| DRO | 61.67 | 51.62 | 56.49 | 64.64 | 51.58 | 53.41 | 53.29 | 51.12 |
| AEDL | 51.38 | 63.31 | 59.89 | 76.36 | 81.35 | 83.41 | 75.51 | 50.62 |
| OpenMAX | 64.98 | 65.37 | 65.54 | 91.21 | 92.26 | 93.38 | 92.53 | 52.62 |
| **AREO** | 72.52 | 74.33 | 72.69 | 92.75 | 96.86 | 96.69 | 96.02 | 55.49 |

## D.7 closed set Performance

Table 10 below shows the closed set performance for competitive baselines. It is interesting to see that DRO with a flexible uncertainty set performs the worst in the closed set setting as it does not learn properly from the most representative samples in the training data while only focusing on the difficult ones. AEDL performs very competitively and achieves the best performances on two datasets. This is partly because we are evaluating MAP by treating the minority class as positive and oversampling helps to improve the prediction on the minority class quite significantly. AREO also performs competitively and achieves the best performance on the other two datasets. The good closed set performance further confirms our theoretical result that proves the equivalence between AREO and AdaBoost, which justifies its strong generalization capability.

Table 10: closed set performance (MAP) on all datasets

| Approach | Cifar10 | Cifar100 | ImageNet | MNIST | AHED |
|---|---|---|---|---|---|
| EDL | $55.39 \pm 3.78$ | $31.80 \pm 2.37$ | $55.84 \pm 1.60$ | $99.58 \pm 0.26$ | $40.48 \pm 2.65$ |
| AEDL | $54.98 \pm 0.63$ | $36.11 \pm 0.08$ | $55.62 \pm 0.58$ | $99.62 \pm 0.23$ | $41.36 \pm 3.98$ |
| DRO | $27.16 \pm 5.94$ | $10.50 \pm 0.25$ | $20.71 \pm 1.00$ | $90.87 \pm 3.89$ | $30.02 \pm 0.93$ |
| **AREO** | $54.65 \pm 1.02$ | $36.44 \pm 0.23$ | $55.31 \pm 1.11$ | $99.88 \pm 0.01$ | $49.68 \pm 1.58$ |

| (a) EDL | (b) DRO | (c) AEDL | (d) AREO |
|---|---|---|---|

Figure 4: OSD performance comparison from imbalanced Cifar10 dataset.

## D.8  SUPERIOR OSD PERFORMANCE OF AREO

Figure 4 provides a deeper insight on the superior OSD performance of AREO than other competitive baselines, including EDL, DRO, and AEDL. Cifar10 is used as an illustrative example and similar patterns are obtained on other datasetts. First, while EDL is able to separate outliers from most samples in the majority classes based on their predicted uncertainty scores, it assigns much higher uncertainty scores to samples from the minority class, making them hard to be separated from the outliers. Second, the uncertainty scores for the majority classes span a wide range, which implies that several (difficult) samples from these classes have also been assigned very high uncertainty scores. If the goal is to ensure that most top-ranked samples are true outliers for effective detection in practice, these highly uncertain close-set samples may significantly affect the detection effectiveness. Third, while oversampling can help to better detect the samples from the minority class, which is indicated by lower uncertainty scores achieved by AEDL, most majority classes become much more uncertain and some of them have even a higher average uncertainty score than the outliers. Furthermore, the uncertainty scores from most classes also span a wide range. Finally, DRO effectively narrows down the range of the uncertainty scores as it allows the model to focus more on the difficult samples. However, it does not effectively bring down the high uncertainty scores of the minority class, either, which is still higher than outliers. Similar to DRO, the proposed AREO also manages to keep the uncertainty scores of data samples from the majority classes low so that even the difficult samples are unlikely to be mis-identified as outliers. Meanwhile, it effectively lowers the uncertainty scores of the minority-class examples so that they can better separated from the outliers.

## E  LINK TO SOURCE CODE

For the source code, please click here.

