# OpenReview forum: "Adaptive Robust Evidential Optimization For Open Set Detection from Imbalanced Data"
_ICLR.cc/2023/Conference — ICLR 2023 poster_

### Official Review · Reviewer_c1ud · 2022-10-19

**Confidence:** 5
**Correctness:** 2
**Technical Novelty And Significance:** 3
**Empirical Novelty And Significance:** 3
**Recommendation:** 6

**Clarity, Quality, Novelty And Reproducibility:**

The paper is well written.  The approach seems novel and for the most part makes sense despite what could be a fatal flaw. The experimental results are impressive, but this reviewer really does not have a good grasp of the MAP metric in this context.  The AUROC is much more intuitive for this reviewer.

In any event, the concern is that the EDL loss function is only composed of the sum of square loss in (3) and is missing the KL term that was used in the EDL paper.  Without the KL term as shown in the original EDL paper, the evidence would go to infinity to minimize the sum of the square loss, meaning that EDL has no way to estimate uncertainty.  The KL regularization term anchors the uncertainty by decreasing evidence when the test samples cannot be predicted well. The problem is that the KL term must be properly annealed so that in can initially learn to discriminate before learning how to calibrate the evidence. If the lack of inclusion of the KL term in (3) was a simple reporting error, the paper still would need to explain how the annealing of its weigh parameter lambda is balanced in relation to opening up the DRO set via eta from epoch to epoch.

On page 4, the statement - "...it provides a low uncertainty mass for minority/difficult samples while remain high (in terms of uncertainty mass) for those open set samples." seem to conflict with an earlier statement that minority/difficult samples are assigned high uncertainty.

**Details Of Ethics Concerns:**

No ethical concerns.

**Strength And Weaknesses:**

The strength of the paper is the that the method appears novel, and the experimental results are impressive.  Furthermore, the paper includes some theoretical linkage to boosting (which this reviewer does not have time to verify).  The weakness of the paper is that the Kullback-Leibler regularization term in EDL seems to be ignored, which makes the results questionable. Also, the composition of the scheduler function as a weighted sum of convex and concave functions is a bit ad-hoc.  The main concern is that the convex functions are parameterized by beta, but this does not seem to be the case for the concave functions.

**Summary Of The Paper:**

The paper enhances evidential deep learning (EDL) for open set detection by incorporating distributionally robust optimization with a scheduler function to boost hard samples to predict while at the same time enhancing the boosting for minority classes so they can be learned.  The new method is termed  adaptive robust evidential optimization (AERO), and experimental results clearly demonstrate the utility of the method.

**Summary Of The Review:**

The paper is very interesting with impressive results for open set detection.  The description of the method neglects a critical regularization term used in EDL, which is one of key component the proposed AERO method builds upon.  Without proper discussion of this term and how it is incorporated into AERO, the results in this paper are suspect.

The authors have addressed my major concerns, I can increase my rating to over the acceptance threshold to 6.  This paper has a lot of merits, and I would have no problem if it appears in the program.  The use of EDL, DRO, and MSF has novelty, but the integration of all three methods does not seem to require any new processing ideas or theory that would bring this paper to a definite accept.

---

> ### Author Response · Authors · 2022-11-18
> **Response to Reviewer c1ud**
>
> **Q1: Use of KL regularization term?**
>
>
> Thank you for pointing out this typo. Sorry for the confusion and this has been fixed in the revised paper. The KL term is indeed used in our implementation, which can be verified through the submitted source code. It can be founded in the source file named augmented\_dro\_edl\_losses.py within the mse\_loss function. For the annealing coefficient $\lambda_t$,  we follow the same decay rate as that of the original evidential deep learning model:  $\lambda_t = \min(1.0, t/10)$, where $t$ is the number of epochs.
>
>
> **Q2: Statement - "...it provides a low uncertainty mass for minority/difficult samples while remain high (in terms of uncertainty mass) for those open set samples." seem to conflict with an earlier statement that minority/difficult samples are assigned high uncertainty**
>
> If directly using evidential learning, it will result in a high uncertainty for the minority/difficult samples because of limited samples to learn from. The proposed model aims to tackle this issue by explicitly focusing on learning from minority class/difficult samples (through adaptive DRO training). This will allow the proposed model to learn adequately from these data samples to reduce their uncertainty mass. This has been confirmed by the outstanding OSD performance of the proposed model.

---

### Official Review · Reviewer_gFy7 · 2022-10-21

**Confidence:** 4
**Correctness:** 3
**Technical Novelty And Significance:** 3
**Empirical Novelty And Significance:** 3
**Recommendation:** 6

**Clarity, Quality, Novelty And Reproducibility:**

Clarity:
The paper is organized well and reads smoothly. However, there were many typos some of which are listed here.
Please proofread the entire paper including the appendix.

Various SVM based techniques  have proposed for OSD --> please correct as "have been proposed"

The grammar in the following sentence is not right. Not sure what the subject of the second part of the setence is starting with "while remain" or (remaining).

… thei predicted uncertainty scores. Should be "their".

This process is known is exploitation … Should be "as exploitation"

It was not clear how (w and \beta) were different than (w' and \beta'). Not mentioned in the text.

Quality/Novelty:

Composite scheduler idea and the way to jointly optimize model parameters and  hyperparameters of the composite scheduler by bilinear optimization and population based training seems to be novel.  The connection between Adaboost and AREO is also interesting.

Reproducibility:

Link to code is provided. However, it is hard to judge whether the results are reproducible without running the code given some key parameters are not discussed (such as P).




**Strength And Weaknesses:**


Strengths:

Evidential learning in the presence of minority classes while acknowledging the potential presence of difficult samples in majority classes is a real-world problem that is often overlooked in most open-set classification and evidential learning papers. On that regard this paper studies a highly significant problem and proposes a scheduling algorithm that adjust uncertainty set size during network training.

The connection between AdaBoost and AREO is quite motivating.

It is good to see that the improvement in performance holds even when different number of minority classes with different sizes are included in experiments. Qualitative analysis ranking sample difficulty by different techniques was also very useful.

Weaknesses:

It is not immediately clear whether the improvement is due to population-based training (PBT) or weighting samples by the proposed composite scheduler. An ablation study would be very useful.  This is a concern because none of the compared techniques seem to be using PBT.

It is also concerning that no run-time comparison is provided between proposed technique and others. PBT would significantly increase run-time and the limited improvement in some datasets may not quite justify this significant increase in run-time.


-----------------
Authors' responses satisfactorily address my main concerns about PBT and run time. If there was a score of 7 I would have upgraded my ranking, regrettably there is none, and I don't think this is an "8" paper.


**Summary Of The Paper:**

The paper proposes an Adaptive Robust Evidential Optimization (AREO) technique to tackle sample uncertainty by evidential learning. The main emphasis in the paper is evidential learning in the presence of minority classes. The paper starts with motivating the need for a tradeoff between distributive robust optimization (which deals with difficult samples in the training set) vs. oversampling (which deals with minority classes). An adaptive evidential learning strategy is proposed that gradually increases the size of the uncertainty set by a composite scheduler function. This function optimizes the weight to be assigned to each training sample at every epoch depending on whether they are from minority classes or majority ones. The optimization involves jointly solving for parameters of the model and the composite scheduler function. A bilinear optimization based on population-based training is proposed. Additional valuable insight about the connection between AREA and AdaBoost is provided under certain assumptions.


**Summary Of The Review:**

The paper studies an important aspect of open-set classification often overlooked in the literature. The proposed idea of using composite scheduler function for adaptive evidential learning is quite intriguing. However, there were some concerns about whether the improvement comes from training P different models at the same time or by optimizing the scheduler function that adjusts the size of the uncertainty set. An ablation study would have been useful to disentangle these two components. It would also be very useful if the weights assigned to different samples during training and evidence computed for open-set samples during testing could have been demonstrated in a simple illustrative set-up.

Questions:

1. It was not also immediately clear what the significance of optimizing \beta would be. SF_m functions can be prefixed,  perhaps a large number of them with different \beta as it is commonly done in composite kernel learning techniques[*]

*Fung, G., Dundar, M., Bi, J., & Rao, B. (2004, July). A fast iterative algorithm for fisher discriminant using heterogeneous kernels. In Proceedings of the twenty-first international conference on Machine learning (p. 40).

2. Training uses the Adam optimizer with a fixed learning rate of 0.001. Neural nets tend to  learn from easy samples (dominant patterns) in earlier epochs and from difficult samples (rare patterns) in later stages. However the fact that no learning rate scheduler is used may not allow enough time for the network to learn from difficult samples, because training may stop prematurely because of the high learning rate without fully exploring the weight space. When comparing the proposed approach against state-of-the-art it may be more convincing if the AREO is also compared against an evidential network where the learning rate is gradually decreased.

---

> ### Author Response · Authors · 2022-11-18
> **Response to Reviewer gFy7**
>
> We would like to thank the reviewer for the valuable comments/suggestions. We summarize our responses as follows.
>
>
>
> **Q1: Ablation study on whether improvement is due to PBT or weighting samples proposed by the composite scheduler.**
>
> Thank you for the suggestion. We would like to first clarify that PBT and multi-scheduler function (MSF) are tightly coupled in the proposed  multi-scheduler learning mechanism.  They work collectively to achieve an optimal adaptive DRO training process. In particular, PBT is used to determine the combination coefficients of atomic scheduler functions, which leads to an optimal MSF. The learned MSF is used to compute the uncertainty set size $\eta_t$, which is then used to assign weights to different data samples. In our ablation study, we have shown that using only atomic scheduler functions under-performs an optimal MSF learned using PBT. Here, we further compare with a MSF using fixed equal weights (instead of learned by PBT) assigned to each atomic scheduler function. Again, AREO shows a clear advantage than a fixed MSF.
>
>
> | Approach | Cifar10 | Cifar+10 | Cifar+50 |
> |----------|---------|----------|----------|
> | W/O PBT     |66.45   | 28.04    | 66.81    |
> | AREO     | 72.48   | 37.14    | 73.87    |
>
> Table 1: OSD (MAP) performance on Cifar10
>
> | Approach | Cifar100 |
> |----------|---------|
> | W/O PBT     | 53.52   |
> | AREO     | 57.52   |
>
> Table 2: OSD (MAP) performance on Cifar100
>
>
> | Approach | MNIST | Noise | MNIST-Noise | Omnigolot |
> |----------|-------|-------|-------------|-----------|
> | W/O PBT      |86.81 | 92.43 | 85.42       | 87.29     |
> | AREO     | 90.80 | 94.24 | 94.18       | 93.80     |
>
> Table 3: OSD (MAP) performance on MNIST
>
> **Q2: Runtime comparison between proposed technique and other techniques?**
>
> Thank you for the suggestion. Given the special structure of PBT, we leverage multiple processors to run multiple models in parallel. As such, the computational overhead is mostly because of collecting the models, running models on the same GPU, and readjusting the hyperparameters and model parameters for the worst performing models. The table below demonstrates the run time (in seconds) of different techniques. As shown, AREO has a slightly higher training time because of the multiple models used by PBT. It should be noted that the same GPU is used for all models in our case and distributing the job on multiple GPUs will further help to effectively reduce the training time, which is common considering the current GPU architecture. Furthermore, there is no additional overhead during the inference process once the best model is picked.
>
> | Approach | MNIST | Cifar10 | Cifar100 |
> |----------|-------|-------|-------------|
> | EDL      |1348.26 | 3082.32 |3279.41       |
> | AEDL     | 1495.32 |3749.07 | 3518.77     |
> | DRO      | 1295.32 | 3111.53 | 3209.52     |
> | AREO     | 2130.75 | 4432.43 | 4668.73     |
>
> Table 4: Runtime comparison
>
> **Q3: Typos.**
>
>
> Thank you for the suggestion. We have updated our draft by correcting the mentioned typos.
>
>
> **Q4: Difference between ($w$ and $\beta$) and ($w^\prime$ and $\beta^\prime$)**
>
>
> $w$ and $\beta$ are mixing weights and scheduler function parameters to control the uncertainty set $\eta_t$ whereas $w\prime$ and $\beta\prime$ are the mixing weights and scheduler function parameters to readjust class specific weights. Please refer to Eq. 8 and Eq. 11 to see the difference.
>
> **Q5:  Comparison of AREO with an evidential network where the learning rate is gradually decreased.**
>
>
> To make the comparison, we gradually decrease the learning rate for the evidential network as training progresses. Specifically, in each iteration, we compute the new learning rate as $l_t = l_0\exp(-\frac{t}{\kappa})$ where $t$ is the current iteration (with 20,000 total iterations in our case) and $\kappa$ is a hyperparameter. The tables below show the performance with different $\kappa$ values compared against AREO. As can be seen, AREO achieves better performance in all cases.
>
> | $\kappa$ | Cifar10 | Cifar+10 | Cifar+50 |
> |----------|---------|----------|----------|
> | 2000     | 65.39   | 28.18    | 66.46    |
> | 4000     | 66.25   | 28.00    | 64.79    |
> | 6000     |65.03  | 29.23    | 66.75    |
> | AREO| 72.48   | 37.14    | 73.87    |
>
> Table 5: OSD (MAP) performance on Cifar10
>
> | $\kappa$ | Cifar100 |
> |----------|---------|
> | 2000     | 52.69   |
> | 4000     | 51.14   |
> | 6000     | 52.71   |
> | AREO | 57.52   |
>
> Table 6: OSD (MAP) performance on Cifar100
>
>
>
> | $\kappa$ | MNIST | Noise | MNIST-Noise | Omnigolot |
> |----------|-------|-------|-------------|-----------|
> | 2000      | 71.74 | 68.10 | 65.40       | 68.46     |
> | 4000     | 70.91 | 66.67 | 65.56       | 65.20     |
> | 6000      | 86.32 | 83.93 | 82.32       | 80.15     |
> |AREO|  90.80 | 94.24 | 94.18       | 93.80     |
>
> Table 7: OSD (MAP) performance on MNIST

---

### Official Review · Reviewer_FxMN · 2022-10-22

**Confidence:** 4
**Correctness:** 4
**Technical Novelty And Significance:** 3
**Empirical Novelty And Significance:** 3
**Recommendation:** 6

**Clarity, Quality, Novelty And Reproducibility:**

In general, the paper is written well and the proposed method makes sense and it has some novelty. Experimental studies support the main claims regarding the proposed method. The authors provide algorithm for the proposed methods, but the source codes are needed to reproduce the experimental results.

**Strength And Weaknesses:**

The main strengths of the paper can be summarized as follows:
i) Despite some Grammar mistakes, the paper is mainly well written.
ii) Although the authors build their proposed method based on the known methods, the resulting method seems effective. Especially, integrating evidential optimization loss in distributionally robust optimization makes sense.
iii) Showing the theoretical connections between the proposed method and AdaBoost is a plus.
iv) The proposed method outperforms other tested methods.
The main weaknesses of the paper can be summarized as follows:
i) There are some missing references related to general open set recognition methods. Especially there are recent successful open set recognition methods that estimate compact class acceptance regions for open set recognition. These methods must be also discussed in the paper and used for experimental comparison.
ii) The authors use mAP scores for assessing accuracy. However, most of the papers reported Area Under the ROC curve (AUC) scores since this is the common setting used for measuring the detection of performance of the unknown samples.  Therefore, please report AUC scores as well.

Minor Issues:
The authors use the term “close set”, but the correct term is the closed set. Please correct it. Also, there are some minor Grammar mistakes that must be corrected (e.g., have been proposed – page , at the beginning of Section 2).

References:
[1] G. Chen, P. Peng, X. Wang, Y. Tian, Adversarial reciprocal points learning for open set recognition, in: arXiv:2103.00953, 2021.
[2] P. Perera, V. I. Morariu, R. Jain, V. Manjunatha, C.Wigington, V. Ordonez, V. M.  Patel, Generative-discriminative feature representations for open-set recognition, in: CVPR, 2020.
[3] H.-M. Yang, X.-Y. Zhang, F. Yin, Q. Yang, C.-L. Liu, Convolutional prototype network for open set recognition, IEEE Transactions on Pattern Analysis and Machine Intelligence (2020) 1–1 doi:10.1109/TPAMI.2020.3045079.
[4] H. Cevikalp, B. Uzun, O. Kopuklu, G. Ozturk, Deep compact polyhedral conic classifier for open and closed set recognition, Pattern Recognition 119 (2021).
[5] A. R. Dhamija, M. Gunther, T. E. Boul, Reducing network agnostophobia, in: Neural Information Processing Systems (NeurIPS), 2018.



**Summary Of The Paper:**

In this paper, the authors focus on open set recognition (detection) and propose a method when the known classes are imbalanced. The proposed method is built based on evidential learning and distributionally robust optimization. To this end, the authors first integrate the evidential learning loss function to distributionally robust optimization and introduce distributionally robust evidential optimization. Then Adaptive Robust Evidential Optimization method is proposed by utilizing multi-schedular function. The main idea is to introduce an optimal training behavior that gives sufficient attention to the difficult samples and minority class and at the same to detect the unknown class samples with high accuracy. The authors show the connection between their proposed method and AdaBoost and compare their method to the related methods. Better accuracies are reported against the rival tested methods.

**Summary Of The Review:**

In general this is a good paper. The proposed method is built based on existing methods, yet it has some novelty. Experimental results support that the proposed method outperforms related methods. However, some important references on open set recognition are missing, the paper also lacks comparison to these methods as well. Also, in addition to mAP scores, AUC scores should be reported as well.

---

> ### Author Response · Authors · 2022-11-18
> **Response to Reviewer FxMN**
>
> We would like to thank the reviewer for the valuable comments/suggestions. We summarize our responses as follows.
>
>
> **Q1: Missing references related to general open set recognition methods.**
>
> Thank you for suggesting these recent general open set recognition methods. We provide an overview of all these approaches in the updated related work section (see the highlighted part in the blue color). We also include a detailed discussion of each of them in the newly added Section D.5.2 in the Appendix.
>
> In addition, we perform a comparison with three methods in references [1], [4], and [5], including two most recent baselines ($i.e.,$ [1] and [4]) with competitive OSD performance. We report the comparison results on Cifar 10, Cifar 100, and MNIST datasets. The tables below demonstrate the average performance taken over 3 runs for each method. The number in parenthesis indicates the paper number mentioned by the reviewer and the columns show the open set testing performance. Since all these methods are general open set recognition models, they are challenged by an imbalanced class distribution, leading to a relatively lower OSD performance as compared with the proposed AREO model in all cases.
>
>
> | Approach | Cifar10 | Cifar+10 | Cifar+50 |
> |----------|---------|----------|----------|
> | ARPL [1]     | 58.90   | 29.33    | 69.00    |
> | Agnostophobia [5]     | 68.50   | 29.11    | 68.36    |
> | DC ECPP  [4]     | 61.22   | 23.54    | 58.30    |
> | AREO     | 72.48   | 37.14    | 73.87    |
>
> Table 1: OSD (MAP) performance on Cifar10
>
> | Approach | Cifar100 |
> |----------|---------|
> | ARPL [1]     | 48.74   |
> | Agnostophobia [5]     | 52.93   |
> | DC ECPP  [4]     | 53.22   |
> | AREO     | 57.52   |
>
> Table 2: OSD (MAP) performance on Cifar100
>
>
>
> | Approach | MNIST | Noise | MNIST-Noise | Omnigolot |
> |----------|-------|-------|-------------|-----------|
> | ARPL [1]      | 90.27 | 74.57 | 50.38       | 39.08     |
> | Agnostophobia [5]     | 85.03 | 78.43 | 72.72       | 75.76     |
> | DC ECPP [4]      | 77.56 | 55.80 | 71.91       | 89.44     |
> | AREO     | 90.80 | 94.24 | 94.18       | 93.80     |
>
> Table 3: OSD (MAP) performance on MNIST
>
>
>
>
>
> **Q2: AUC Scores**
>
> Thank you for the suggestions. The tables below report the AUC scores for multiple competitive baselines on three datasets. The AUC scores present a trend that is consistent as that of the MAP scores. These results have been also included in Table 9 of Appendix D.6 in the revised paper.
>
>
> | Approach | Cifar10 | Cifar+10 | Cifar+50 |
> |----------|---------|----------|----------|
> | EDL     | 66.36   | 70.66    | 69.07    |
> | DRO     | 61.67   | 51.62    | 56.49    |
> | AEDL     |51.38   | 63.31    | 59.89    |
> | OpenMAX     | 64.98   | 65.37    | 65.54    |
> | AREO     | 72.52   | 74.33   | 72.69    |
>
> Table 4: OSD (AUC) performance on Cifar10
>
> | Approach | Cifar100 |
> |----------|---------|
> |EDL     | 52.38   |
> | DRO     | 51.12   |
> | AEDL     | 50.62   |
> | OpenMAX     | 52.62   |
> | AREO | 55.49 |
>
> Table 5: OSD (AUC) performance on Cifar100
>
>
>
> | Approach | MNIST | Noise | MNIST-Noise | Omnigolot |
> |----------|-------|-------|-------------|-----------|
> | EDL      | 88.72 | 95.36 | 92.31       | 91.23     |
> | DRO     | 64.64 | 51.58 | 53.41       | 53.29    |
> | AEDL      | 76.36 |81.35 | 83.41       | 75.51     |
> | OpenMAX      | 91.21 | 92.96 | 93.38       | 92.53     |
> | AREO     | 92.75 | 96.86 | 96.69       | 96.02     |
>
> Table 6: OSD (AUC) performance on MNIST
>
>
>
> **Q3: Grammatical errors and typos**
>
>
> Thank you for identifying these presentation issues. We have fixed those typos and grammatical errors in the updated draft. A careful proofreading will also be conducted to further improve the quality of the presentation.

---

### Official Review · Reviewer_HU7D · 2022-10-25

**Confidence:** 2
**Correctness:** 3
**Technical Novelty And Significance:** 2
**Empirical Novelty And Significance:** 2
**Recommendation:** 6

**Clarity, Quality, Novelty And Reproducibility:**

The paper is generally well written.

Below Eq.6: $\eta$ could be defined.  The uncertainty set seems to be central to the approach, a clear description of it would be helpful.

The novelty is not high because the proposed AREO is a combination of existing approaches.

**Strength And Weaknesses:**

Strengths:

1.  The problem of OSD with imbalanced data is interesting

2.  The empirical results indicated the proposed AREO compares favorably with 8 algorithms and 5 datasets.

Weaknesses:

1.  The proposed AREO is a combination of  the existing approaches DRO and EDL, with MSF (multi-scheduler function), which also is not new.


**Summary Of The Paper:**

The authors address the problem of imbalanced data in open set detection (OSD) and proposed AERO (Adaptive Robust Evidential Optimization).  Based on evidential deep learning (EDL), their approach includes an uncertainty mass for the unknown class, which is used for detecting the unknown class.  Also, they use DRO (Distributionally Robust Optimization) to handle imbalanced data via learning weights for each instances to focus on harder instances.    They propose distributionally robust evidential loss (DREL), which is a combination of DRO and EDL.  They replace the loss function for each instance in DRO with the loss function from EDL.  To learn from easy to hard instances (hence not focusing only on hard instances), they propose MSF (mutli-scheduler function) to adaptively increase the size of uncertainty set.  MSF is a weighted sum of multiple scheduler functions, and the weights are learned.  Adaptive Robust Evidential Loss (AREL) adds an adaptive size of the uncertainty set $\eta$ according to MSF to DREL.   To increase the chance of minority instances to be included in the uncertainty set (in additional to hard majority instances), they increase the weights for the minority instances.  During training, they alternate between AREL loss and MSF loss.  In their theoretical analysis, they link their approach to AdaBoost.

They compare their proposed AREO with 8 existing algorithms and 5 datasets.  Their results indicate AREO compares favorably.


**Summary Of The Review:**

The paper addresses an interesting problem in OSD with imbalanced data.  The authors proposed combining existing EDL (for OSD) and DRO (for imbalanced data) with MSF (for adaptively increasing the uncertainty set).

---

> ### Author Response · Authors · 2022-11-18
> **Response to Reviewer HU7D**
>
> We would like to thank the reviewer for the valuable comments/suggestions. We summarize our responses as follows.
>
>
> **Q1: Clarification on novelty.**
>
> Thank you for your comment. We would like to first explain why directly using existing techniques, including EDL and DRO, will fail to address the key challenges for open set detection (OSD) from imbalanced data. We then clarify how the proposed AREO (Adaptive Robust Evidential Optimization) makes novel and nontrivial extensions to existing techniques. The effectiveness of AREO has also been justified through our novel theoretical analysis and comprehensive experiments.
>
> - First, due to the imbalanced class distribution, evidential learning may not learn properly from the minority classes due to lack of data samples. As a result, it will be confused between the minority class samples and those from the open set, leading to a rather poor OSD performance, as evidenced by our empirical evaluation as shown in Table 1, where EDL is usually more than 10\% worse than the proposed AREO model.
>
> - Second, directly applying DRO with a flexible uncertainty set may put too much emphasis on the difficult samples, which makes it ignore the minority class as well as some representative samples from the majority classes. This can significantly affect proper model training, leading to a much worse OSD performance that is usually more than 20\% worse than the proposed AREO model.
>
> A fundamental challenge of our problem setting lies in the interplay between samples from the minority class and the difficult samples from the majority classes. The poor OSD performance from EDL or DRO alone clearly indicates that a carefully designed training process is needed to ensure an optimally balanced training behavior that gives sufficient attention to the difficult samples and the minority class while capable of learning common patterns from the majority classes. While the uncertainty set in DRO allows us control how much attention to put on the difficult samples, it does not offer an optimal training schedule as required and provides no guarantee to ensure adequate attention to the minority classes. The proposed adaptive DRO training seamlessly integrates a novel multi-scheduler learning mechanism and ratio biased weight augmentation to achieve an optimal training schedule through bi-level optimization, which leads to significantly improved OSD performance as shown in Table 1. We believe that the multi-scheduler learning mechanism, which is essential to achieve an optimal adaptive model training behavior, is a novel contribution. It leverages a uniquely designed multi-scheduler function (MSF) with rich expressive power to capture the complex training behavior of different datasets and automatically optimizes the MSF through bi-level optimization. Schedulers have been primarily used in the context of distributed/parallel computing to manage the computation workload, which is fundamentally different from our problem setting. Simple scheduler functions have also been used in curriculum learning but lacks sufficient expressiveness and adaptiveness to accommodate a much more complex training behavior required by open set detection from imbalanced data. Finally, we also theoretically show the connection between AREO with  AdaBoost, which ensures its nice convergence and generalization properties.
>
> **Q2:  Below Eq.6: $\eta$ could be defined. The uncertainty set seems to be central to the approach, a clear description of it would be helpful.**
>
>
> Thank you for the suggestion. We have added a more detailed description on the role of $\eta$ right after Eq. (5), when $\eta$ is first introduced (highlighted in blue in the revised paper). In particular, when  $\eta$ is large, the weight distribution ${\bf p}$ can deviate a lot from the uniform distribution, making it possible to assign a very high weight to certain data samples. In contrast, a small $\eta$ will constrain ${\bf p}$ to be close the uniform distribution and all samples share a similar weight.

---

> > ### Comment · Reviewer_HU7D · 2022-11-24
> > **comment on author response**
> >
> > Thanks for the two bullet points on the limitations of EDL and DRO.  I suggest you to put them together in one discussion in the paper to clarify either one is insufficient and why a scheduler to combine them could help.   Also, in the discussion, I suggest to include evidence for "confused between the minority class samples and those from the open set" and "put too much emphasis on the difficult samples, which makes it ignore the minority class as well as some representative samples from the majority classes".
> >
> > Thanks for further explanation on $\eta$.

---

> > > ### Author Response · Authors · 2022-11-25
> > > **Thank you for the suggestions**
> > >
> > > Thank you for the suggestions! We will make sure to include a discussion in the revised paper that covers both bullet points to clarify the limitations of EDL and DRO and highlight why a scheduler to combine them is essential.
> > >
> > > As for the evidence to justify the limitations of EDL and DRO, the main OSD performance in Table 1 provides the overall evidence, showing that using EDL and DRO directly under-perform the proposed AREO model by a large margin (around 10\% and 20\%, respectively, on average). Figure 4 in Appendix D.8 provides more direct evidence to demonstrate the superior OSD performance of AREO and the limitations of other models, including EDL and DRO. As can be seen in Figure 4 (a),  EDL assigns high uncertainty scores to samples in the minority class ($i.e.,$ bird), making it confused between the minority class samples and those from the open set (because both have high uncertainty scores). As for DRO, Figure 4 (b) shows that it effectively narrows down the range of the uncertainty scores as it allows the model to focus more on the difficult samples. However, it does not effectively bring down the high uncertainty scores of the minority class, either, which is still higher than outliers. Meanwhile, some majority class ($e.g.,$ airplane) samples have been assigned higher uncertainty scores as well. As a result, DRO will wrongly detect many samples from both  minority and majority classes as open-set ones. We will refer to these results as evidence in our added discussion.

---

> > > > ### Comment · Reviewer_HU7D · 2022-11-29
> > > > **comment on author response**
> > > >
> > > > Please indicate where the updates appear in the revised paper.
> > > >
> > > > On evidence of the limitations of EDL and DRO, please add in the paper more "direct" evidence to justify:
> > > >
> > > >     "confused between the minority class samples and those from the open set" and
> > > >
> > > >    "put too much emphasis on the difficult samples, which makes it ignore the minority class as well as some representative samples from the majority classes".
> > > >
> > > > That is, not overall performance comparison, but justification for your claimed limitations.   If space is an issue, you can discuss your evidence/justification and reference sections in the appendix.

---

> > > > > ### Author Response · Authors · 2022-11-30
> > > > > **Thank you for these follow-up questions**
> > > > >
> > > > > Thank you for these follow-up questions! First, we apologize for the confusion in our previous response. Since we were not allowed to further change the draft after the Nov 18 deadline, we meant to include the discussion that covers both bullet points in the final version of the paper (if it can be accepted).
> > > > >
> > > > > As for showing evidence on the limitations of EDL and DRO, besides Figure 4 and corresponding discussions in Appendix D.8, our qualitative analysis and illustrative examples shown in  Figure 2 provide additional direct evidence. More specifically, in Figure 2 (a), the first row shows the minority-class samples and Figure 2 (b) shows the ranking of these samples according to the uncertainty scores, where a lower number indicates a lower uncertainty. In the case of EDL, the model is confused between minority-class samples and open set ones and this is evidenced by the high uncertainty scores assigned to the minority-class samples. In the case of DRO, as it focuses on the most difficult samples, it misses minority-class samples as well as many representative samples from majority classes. This can be observed in Figure 2 (b), where DRO assigns high uncertainty scores to minority-class and representative majority-class samples. In contrast to EDL and DRO, AREO assigns much lower uncertainty scores to these samples from known classes. For more details, please also refer to Section 4.4 of the main paper.

---

> > > > > > ### Comment · Reviewer_HU7D · 2022-12-05
> > > > > > **comment on author response**
> > > > > >
> > > > > > Please elaborate on why a scheduler combining them could overcome the limitations of EDL And DRO.

---

> > > > > > > ### Author Response · Authors · 2022-12-06
> > > > > > > **Elaboration on the multi-scheduler function**
> > > > > > >
> > > > > > > Thank you for the the follow-up question. By using the proposed multi-scheduler function (MSF), AREO will first put equal attention on all the data samples to learn the general and representative patterns from the entire  data space. This is achieved by a relatively small uncertainty set size $\eta_t$ realized by a large MSF value (see Eq. 7).  As training progresses, the MSF value gradually decreases and $\eta_t$ will increase accordingly, where the rate of change depends on the nature of dataset, which is automatically optimized through adaptive robust training. Furthermore, to make model explicitly focus on the minority classes, we perform ratio biased weight augmentation, which also leverages the proposed MSF (see Eq. 10). As the MSF value gradually approaches zero, the total weight for the minority class samples will eventually reach to $\frac{1}{C}$, making the minority class equally weighted as other majority classes.

---

> > > > > > > > ### Comment · Reviewer_HU7D · 2022-12-10
> > > > > > > > **comment on author response**
> > > > > > > >
> > > > > > > > Limitations:
> > > > > > > >
> > > > > > > > 1. "confused between the minority class samples and those from the open set" and
> > > > > > > >
> > > > > > > > 2. "put too much emphasis on the difficult samples, which makes it ignore the minority class as well as some representative samples from the majority classes".
> > > > > > > >
> > > > > > > > The scheduler seems to address "ignore the minority class" part in the second limitation.  Please elaborate on why the scheduler can address the first limitation and the "some representative samples from the majority classes" in the second limitation.

---

> > > > > > > > > ### Author Response · Authors · 2022-12-11
> > > > > > > > > **Elaborate on why the scheduler can address the first limitation**
> > > > > > > > >
> > > > > > > > > Thank you for the question. As per Eq. 7, during the earlier phase of training, a large MSF value will be assigned, which makes the uncertainty set size $\eta_t$ small. According to Eq. 4, small $\eta_t$ constrains the sample weights to be close the uniform distribution, so all samples share approximately equal weights. As a result, the model will place similar attention to all samples, which allows it to learn the common patterns captured by the representative data samples from the majority classes. As training progresses, the model will start to focus on the difficult samples (by adaptively adjusting the MSF value), where the model will fine-tune its general knowledge by learning from the more challenging samples in the majority classes as well as the samples from the minority classes as explained in our previous response.

---

### Decision · Program_Chairs · 2023-01-20

**Decision:**

Accept: poster

**Justification For Why Not Higher Score:**

The scope is quite focused (imbalanced data in open set detection).

**Justification For Why Not Lower Score:**

All reviewers found the topic interesting.
Although the paper has incremental novelty (combining some methods), the authors justify their approach and make clear their contributions of their proposed AREO and theoretical analysis.

**Metareview: Summary, Strengths And Weaknesses:**

**Summary:** the paper focuses on open set recognition (detection) and proposes a method when the known classes are imbalanced. The proposed method is built based on evidential learning and distributionally robust optimization. To this end, the authors first integrate the evidential learning loss function to distributionally robust optimization and introduce distributionally robust evidential optimization. Then Adaptive Robust Evidential Optimization method is proposed by utilizing multi-schedular function. The main idea is to introduce an optimal training behavior that gives sufficient attention to the difficult samples and minority class and at the same to detect the unknown class samples with high accuracy. The authors show the connection between their proposed method and AdaBoost and compare their method to the related methods. Better accuracies are reported against the rival tested methods.

**Strengths**: well written; although built on previous methods, the combination is effective. some theoretical connections with AdaBoost. Good performance compared with other tested methods.

**Weaknesses**: The following weaknesses were identified by reviewers:

1. incremental novelty, combination of existing approaches (DRO, EDL & MSF). [HU7D]
2. missing refs in related works and experiments. [FxMN]
3. use AUC for evaluation instead of mAP. [FxMN, c1ud]
4. missing ablation study on PBT or weighted sampling of the scheduler [gFy7]
5. missing run-time comparison [gFy7]
6. significance of optimizing beta in the scheduler? [gFy7]
7. also compare against learning rate decay [gFy7]
8. KL divergence term in EDL is missing [c1ud]
9. how to set the parameter lambda in EDL [c1ud]
10. adhoc nature of the composite scheduler, beta does not apply to concave functions? [c1ud]
11. various minor typos

**Discussion:**
The authors wrote a response to address these concerns. During the discussion, all reviewers were satisfied with the response. Reviewer FxMN noted that:
  - However, I have some doubts regarding the latest results comparing their proposed method to recent SOTA open set recognition methods. Although SOTA open set recognition methods do not have any mechanism for handling imbalanced datasets, I would not expect such bad accuracies since only one class is imbalanced and it still has 30% of original sample size. Therefore, the authors should double check these results.
  - There are still minor typos that must be corrected (e.g., although the authors replaced the close set with closed set in some places, there are still many close set terms in the paper).

In the end, the final rating was 6666. Two reviewers would have liked to give a "7" if possible, thinking that the paper is not strong enough for "8".   The paper can be published after further minor revision according to the discussion and feedback.

**Note From Pc:**

if the above contains the word "oral" or "spotlight" please see: "oral" presentation means -> notable-top-5% and "spotlight" means -> notable-top-25%. As stated in our emails, we are disassociating presentation type from AC recommendations

**Summary Of Ac-Reviewer Meeting:**

n/a